

# A measurement-based verification framework for UK greenhouse gas emissions: an overview of the Greenhouse gAs Uk and Global Emissions (GAUGE) project

Paul I. Palmer[1], Simon O'Doherty[2], Grant Allen[3], Keith Bower[3], Hartmut Bösch[4], Martyn P. Chipperfield[5], Sarah Connors[7], Sandip Dhomse[6], Liang Feng[1,8], Douglas P. Finch[1], Martin W. Gallagher[3], Emanuel Gloor[6], Siegfried Gonzi[1,9], Neil R. P. Harris[10], Carole Helfter[11], Neil Humpage[4], Brian Kerridge[12,13], Diane Knappett[12,13], Roderic L. Jones[7], Michael le Breton[3,14], Mark F. Lunt[2], Alistair J. Manning[15], Stephan Matthiesen[1], Jennifer B.A. Muller[3,16], Neil Mullinger[11], Eiko Nemitz[11], Sebastian O'Shea[3], Robert J. Parker[4], Carl J Percival[3,17], Joseph Pitt[3], Stuart N. Riddick[7], Matthew Rigby[2], Harjinder Sembhi[4], Richard Siddans[12,13], Robert L. Skelton[7], Paul Smith[7,18], Hannah Sonderfeld[4], Kieron Stanley[2], Ann R. Stavert[2], Angelina Wenger[2], Emily White[2], Christopher Wilson[5,19], and Dickon Young[2]

[1]School of GeoSciences, University of Edinburgh, UK
[2]School of Chemistry, University of Bristol, UK
[3]Centre for Atmospheric Science, The University of Manchester, Manchester, UK
[4]Earth Observation Science Group, Department of Physics and Astronomy, University of Leicester, Leicester, UK
[5]School of Earth and Environment, University of Leeds, Leeds, UK
[6]School of Geography, University of Leeds, Leeds, UK
[7]Centre for Atmospheric Science, University of Cambridge, Cambridge, UK
[8]National Centre for Earth Observation, University of Edinburgh
[9]Now at the Met Office, Exeter, UK
[10]Centre for Environmental and Agricultural Informatics, Cranfield University, Cranfield, UK
[11]Centre for Ecology and Hydrology, Penicuik, UK
[12]Space Science and Technology Department, Rutherford Appleton Laboratory, Oxfordshire, UK
[13]National Centre for Earth Observation, Rutherford Appleton Laboratory, UK
[14]Now at Department of Chemistry & Molecular Biology, University of Gothenburg, Gothenburg, Sweden
[15]Met Office, Exeter, UK
[16]Now at Deutscher Wetterdienst, Meteorologisches Observatorium Hohenpeißenberg, Hohenpeißenberg, Germany
[17]Now at the Jet Propulsion Laboratory, Pasadena, CA, USA
[18]Now at Institute of Physical Chemistry Rocasolano, Madrid, Spain
[19]National Centre for Earth Observation, University of Leeds, UK

*Correspondence to:* P. I. Palmer
(paul.palmer@ed.ac.uk)





**Abstract.** We describe the motivation, design, and execution of the Greenhouse gAs Uk and Global Emissions (GAUGE) project. The overarching scientific objective of GAUGE was to use atmospheric data to estimate the magnitude, distribution, and uncertainty of the UK greenhouse gas (GHG, defined here as $CO_2$, $CH_4$, and $N_2O$) budget, 2013-2015. To address this objective we established a multi-year and interlinked measurement and data analysis programme, building on an established tall tower GHG measurement network. The inter-calibrated measurement network comprises ground-based, airborne, ship-borne, balloon-borne, and space-borne GHG sensors. Our choice of measurement technologies and measurement locations reflects the heterogeneity of UK GHG sources that range from small point sources such as landfills to large, diffuse sources such as agriculture. Atmospheric mole fraction data collected at the tall towers and on the ships provide information on sub-continental fluxes, representing the backbone to the GAUGE network. Additional spatial and temporal details of GHG fluxes over East Anglia were inferred from data collected by a regional network. Data collected during aircraft flights were used to study the transport of GHGs on local and regional scales. We purposely integrated new sensor and platform technologies into the GAUGE network, allowing us to lay the foundations of a strengthened UK capability to verify national GHG emissions beyond the project lifetime. For example, current satellites provide sparse and seasonally uneven sampling over the UK mainly because of its geographical size and cloud cover. This situation will improve with new and future satellite instruments, e.g. measurements of $CH_4$ from the TROPOMI instrument aboard Sentinel-5P. We use global, nested, and regional atmospheric transport models and inverse methods to infer geographically resolved $CO_2$ and $CH_4$ fluxes. This multi-model approach allows us to study model spread in *a posteriori* flux estimates. These models are used to determine the relative importance of different measurements to infer the UK GHG budget. Attributing observed GHG variations to specific sources is a major challenge. Within a UK-wide spatial context we used two approaches: 1) $\Delta^{14}CO_2$ and other relevant isotopologues (e.g. $\delta^{13}C_{CH4}$) from collected air samples to quantify the contribution from fossil fuel combustion and other sources; 2) geographical separation of individual sources, e.g. agriculture, using a high-density measurement network. Neither of these represents a definitive approach, but they will provide invaluable information about GHG source attribution when they are adopted as part of a more comprehensive, long-term national GHG measurement programme. We also conducted a number of case studies, including an instrumented landfill experiment that provided a test-bed for new technologies and flux estimation methods. We anticipate that results from the GAUGE project will help inform other countries on how to use atmospheric data to quantify their nationally determined contributions to the Paris Agreement.



## 1 Introduction

Human-driven emissions of carbon dioxide ($CO_2$), methane ($CH_4$), nitrous oxide ($N_2O$), and other greenhouse gases (GHGs) to the Earth's atmosphere perturb the balance between net incoming solar radiation and outgoing terrestrial radiation. These emissions, primarily from the combustion of fossil fuels and land-use change activities, are the dominant cause of the warming trend in the climate system since the 1950s (IPCC, 2013). Minimizing the manifold impacts of increasing atmo-

spheric GHGs demands a structured timetable of emission reductions. Avoiding the two-degree Celsius global temperature rise (Nordhaus, 1977) requires that we are already close to peak emissions, with stringent reductions that lead to zero or negative net emissions by 2100. At the Paris Conference of the Parties (COP) in December 2015, 195 countries agreed to accelerate this schedule in order to achieve net zero emissions later this century. Achieving this objective demands accurate

knowledge of national GHG emissions and the contributions from individual sectors. The United National Framework Convention on Climate Change (UNFCCC) requires that all countries included in Annex 1 of that Convention report their annual GHG inventory, including $CO_2$, $CH_4$, and $N_2O$. The bottom-up approach to determining these emissions from individual sectors is on a production, in-use, and disposal basis using source-dependent activity data and emissions factors. A complemen-

tary top-down approach is to verify nationwide GHG emissions using atmospheric measurements of these GHGs, but in practice this is non-trivial and presents many scientific challenges. Here, we describe the UK Natural Environment Research Council (NERC) Greenhouse gAs Uk and Global Emissions (GAUGE) project. In particular, we 1) define the scientific objectives of GAUGE; 2) describe individual measurement types and the atmospheric transport models used to interpret these

data; and 3) outline the broader modelling approach that is adopted in order to determine the magnitude and uncertainty of UK flux estimates of GHGs. Throughout this paper, where relevant, we refer the reader to peer-reviewed publications describing the analysis of individual GAUGE datasets.

The UK Climate Change Act 2008 commits the UK to reduce GHG emissions by at least 80% below 1990 baseline levels by 2050, with an interim target of a 34% reduction compared the same

baseline by 2020. To establish a realistic trajectory towards the 2020 and 2050 goals, the Climate Change Act established five five-year carbon budgets (2008−2032). Seven GHGs are the subject of these staged emission reductions: $CO_2$, $CH_4$, $N_2O$, hydrofluorocarbons, perfluorocarbons, sulphur hexafluoride, and nitrogen trifluoride.

UK government statistics report that $CO_2$, $CH_4$, and $N_2O$ correspond to $\simeq 81$%, 11%, and 5% of

the estimated UK 495.7 MtCO$_2$e (budget in 2015, Department for Business Energy and Industrial Strategy (2017)); the remaining 3% is due to fluorinated gases. This budget, broken down by sector in 2015: energy supply (29%), transport (24%), business (17%), residential (13%), agriculture (10%), waste management (4%), industrial processes (2%), and other (1%). Emissions of $CO_2$ are largest for energy supply, transport, business, and residential sectors. $CH_4$ emissions are largest for agricul-

ture and waste management, and $N_2O$ emissions are largest for agriculture. These emission sources



are very different in nature, ranging from point sources (e.g. industry) to geographically large, diffuse sources (e.g. agriculture). We take into account these differences in the GAUGE measurement strategy, as described below.

The primary objective of GAUGE is to quantify the magnitude, distribution, and uncertainty of the UK GHG $CO_2$, $CH_4$, and $N_2O$ budgets, 2013–2015. Our rationale is that better understanding the national GHG budget will inform the development of effective emission reduction policies that help the UK to meet the interim targets of the UK Climate Change Act and to achieve its commitments to the Paris Agreement. To achieve our primary objective we put together a 42-month research programme, bringing together a purpose-built atmospheric measurement network and a range of atmospheric transport models and inverse methods to translate those measurements into UK GHG flux estimates. More broadly, GAUGE provides an assessment of our current ability to infer GHG fluxes from atmospheric data, and strengthens the UK capability to verify national GHG budgets beyond the lifetime of GAUGE.

GAUGE builds on a long heritage of UK atmospheric observations that have been used to estimate national GHG emissions. Manning et al. (2003) were the first to apply an inverse model approach to infer UK $CH_4$ and $N_2O$ emissions, using data collected from Mace Head (MHD), Ireland, during 1995–2000. This approach contrasted clean upwind air that arrived from the North Atlantic with air masses that passed over mainland UK and Europe and influenced by continental fluxes (Villani et al., 2010). Although, these data provided incomplete measurement coverage of the UK, results using this method have been part of the UK reporting to the UNFCCC. In later work, Polson et al. (2011) used research aircraft observations of GHG mole fractions from the NERC-funded AMPEP campaign (Aircraft Measurement of Chemical Processing and Export fluxes of Pollutants over the UK) to infer fluxes of $CO_2$, $CH_4$, and $N_2O$ and a range of halocarbons. During AMPEP the research aircraft circumnavigated the UK during the summer of 2005 and September 2006. They found that the inferred $CO_2$ fluxes during the campaign were close to the bottom-up emission inventory, but $CH_4$ and $N_2O$ fluxes were much larger than the inventory data but with significant uncertainties. The main advantage of using an aircraft is its ability to sample nationwide scale emissions over a relatively short time period. However limited sorties during AMPEP left gaps in sampling, which affected their ability to describe GHG emissions that include large seasonal cycles (e.g. agriculture).

For more than a decade the UK has included a verification annex chapter to its annual National Inventory Report to the UNFCCC (https://www.unfccc.int). This chapter provides an annual comparison of the reported GreenHouse Gas Inventory (GHGI) of each reported gas to those estimated using atmospheric observations and the Bayesian inverse modelling technique InTEM (Inversion Technique for Emission Modelling). The precursor to InTEM is described by Manning et al. (2011). InTEM uses the output from the NAME (Numerical Atmospheric dispersion Modelling Environment) transport model (Manning et al., 2011), which describes how emissions disperse and dilute in the atmosphere, and observations from the UK DECC (Deriving Emissions related to Climate Change)





tall tower network (described below). A recent study used NAME and a hierarchical Bayesian approach to determined UK emissions of $CH_4$ and $N_2O$ using the UK DECC network from 2012 to 2014 (Ganesan et al., 2015). They found that *a posteriori* fluxes were lower than *a priori* values. Using geographical distributions of sectoral emissions, Ganesan et al. (2015) tentatively attributed their result to an overestimation of agricultural emissions of $CH_4$, and a significant seasonal cycle of $N_2O$ emissions. Recent work has incorporated the reversible jump Markov Chain Monte Carlo (MCMC) inverse modelling method (Lunt et al., 2016). The main advantage of this new approach is that the algorithm chooses the number of the unknown parameters, including the geographical size of the region, to be solved given the data. *A posteriori* $CH_4$ emissions for March 2014 inferred from the DECC network data were consistent with Ganesan et al. (2015) (Lunt et al., 2016). Within the GAUGE project InTEM is used together with other inverse methods (section 3) to provide an ensemble of flux estimates, which provide a broader picture of the range of estimates. Using InTEM also provides a link between GAUGE and previous UK GHG estimates.

The measurement strategy we have adopted within GAUGE includes long-term measurements and shorter-term, higher-resolution network measurements, focused aircraft experiments, $CO_2$ sondes, characterization of point sources such as landfills, and satellite remote sensing. Our approach accounts for the heteorogeneity of UK sources, e.g. point sources for power generation to large, diffuse and seasonal sources from agriculture. It also addresses the need to focus attention on smaller regional and city scales. This focus on smaller regions will progressively grow in importance with ongoing rapid rates of urbanization across the world. GAUGE included new *in situ* and remote sensing technologies, and new measurement platforms (e.g. unmanned aerial vehicles) that will help to future-proof the UK GHG measurement network. To help attribute observed variations in atmospheric GHGs to individual sources, e.g. fossil fuel combustion, we explored the potential of isotopologues to chemically identify source signatures, and high-density measurements to exploit geographical distributions of individual sector emissions.

In section 2 we describe the measurements we collected during GAUGE and the attributes that make them ideal for quantifying nationwide GHG fluxes. We also discuss the inter-calibration efforts that put these different data on internationally-recognized calibration scales, placing GAUGE data into a wider context. In section 3 we describe the models we use to describe atmospheric chemistry and transport, the challenges faced, and the associated inverse methods that we use to infer GHG fluxes from the GAUGE data. We conclude in section 4.

## 2 Measurements

We present an overview of the measurements collected as part of GAUGE in Tables 1, 2, 4, 5, 6, and 8. We distinguish between *in situ* measurements, mobile measurements platforms, and space-borne data. We also include a description of how we inter-calibrate these different data.





### 2.1 In Situ Measurements

We use tall tower measurements and the atmospheric baseline observatory at MHD to provide a
long-term *in situ* measurement record to underpin the main objectives of GAUGE. Tall towers are
used to collect atmospheric GHG measurements that are sensitive to fluxes on a horizontal scale
of 10–100s km. We also established a geographically dense network of observations to help isolate
GHG emissions from individual sources.

### Tall Tower Measurement Network

Figure 1 shows the geographical locations of the tall towers (TTs) that collect atmospheric measurements of GHGs (Tables 1 and 2) and provide the long-term, core measurement capability of the
UK GHG measurement network. Sampling air high above the land surface reduces the influence
of local signals that can compromise interpretation of observed variations of GHGs (Gerbig et al.,
2003, 2009). With the exception of the MHD atmospheric research station (described below) air is
typically sampled at least 50 m above the local terrain and at multiple heights (Table 1) to assess the
role of atmospheric mixing in the planetary boundary layer.

Tables 1 and 2 describe the five TT locations and the MHD site used in the GAUGE project.
High-frequency measurements of GHGs have been collected for the past three decades at the MHD
northern hemisphere background measurement station on the west coast of Ireland. They predominately represent clean, western baseline conditions for the UK and mainland Europe. These MHD
data have been previously used to infer UK-wide GHG emissions (Manning et al., 2011). In 2012,
the UK Deriving Emissions linked to Climate Climate (UK DECC) tall tower network was established across mainland UK using funding from the UK Department of Energy and Climate Change
(with the responsibility now residing in the Department for Business, Energy and Industrial Strategy,
BEIS). Three sites were established (Angus, Ridge Hill, and Tacolneston, Table 1) with the purpose
of improving the spatial and temporal distribution of measurements across the UK to reduce uncertainties of GHG emissions for the devolved administrations (i.e. England, Wales, Scotland, and
Northern Ireland). As part of the GAUGE project, we augmented the UK DECC network with two
TT sites at Bilsdale and Heathfield (Figure 1) that started collecting data from 2013 onwards. These
two new sites were chosen to help fill the measurement coverage over mid-northern England, where
there is significant industrial activity, and to collect measurements south of London. For detailed
descriptions of each site, measurement and data logging instrumentation, and the calibration protocols we refer the reader to Appendix A, Stanley et al. (2017) and A. R. Stavert et al, "GAUGE Tall
Towers: measurements, methodologies and impact," in preparaton for *Atmos. Chem. Phys. Discuss.*,
2018 - hereafter ARS18a.

As an example, Figure 2 shows $CO_2$, $CH_4$, and $N_2O$ mole fraction data from Bilsdale, North
Yorkshire. Figure 2 also shows the statistically determined baseline, long term trend and mean diur-





nal cycle for each season. The statistical fitting procedure is decribed in Thoning et al. (1989), and

on the associated NOAA/ESRL website http://www.esrl.noaa.gov/gmd/ccgg/mbl/crvfit/crvfit.html. The mean Bilsdale growth rates for $CO_2$, $CH_4$ and $N_2O$ are 3 ppm/yr, 8 ppb/yr and 0.8 ppb/yr, respectively. The mean seasonal amplitudes for these gases are 18 ppm, 51 ppb, and 0.8 ppb, respectively. Table 3 summarizes the descriptive statistics for tall towers data. Diurnal variations of these gases vary seasonally, particularly $CO_2$ and $CH_4$ that have large surface fluxes. Fluxes of $CO_2$, for instance, have a peak diurnal cycle of $\simeq$10 ppm during summer months. Diurnal variations during

winter months, particularly evident at lower inlet heights, provide some indication of the role of boundary layer height. Variations of $CH_4$ are due to changes in anthropogenic emissions but also to higher summertime OH concentrations, which represent the main loss term. $N_2O$ has an atmospheric lifetime $\simeq$120 years, determined by stratospheric photolysis. Our measurements show a growth rate that is consistent with the global value of $\simeq$0.9 ppb/yr.

We also analyzed the radiocarbon content of $CO_2$ ($\Delta^{14}CO_2$) at MHD and TAC as an approach to estimate the fossil fuel contribution to observed atmospheric variations of $CO_2$ (ff$CO_2$). The underlying idea is that fossil fuels, by virtue of their age, are devoid of $^{14}C$, which has a half-life of 5700±30 years (Roberts and Southon, 2007). Measurements of $\Delta^{14}CO_2$ have been used extensively to determine ff$CO_2$ (e.g. Meijer et al. (1996); Levin et al. (2003); Levin and Karstens

(2007); Turnbull et al. (2006, 2009); Graven et al. (2009); Berhanu et al. (2017)). Our sampling strategy at MHD (nominally unpolluted site) and TAC (nominally polluted site) was designed to determine the west-east gradient of ff$CO_2$, reflecting the prevailing wind direction over the UK.

Weekly glass flask sample pairs were collected at MHD and TAC. A commercial sampling package is used at MHD (Hermes PFP, High Precision Devices Inc., USA) as part of the National Oceanic

& Atmospheric Administration (NOAA) Carbon Cycle Greenhouse Gases global flask sampling program run by the Earth System Research Laboratory (ESRL). Flask pairs have been filled at MHD for NOAA since 1991, but they have not been previously analysed for $^{14}CO_2$. We added an extra flask to the collection from June 2014.

Weekly sampling commenced in June 2014 and concluded in February 2016. To determine the

radiocarbon $CO_2$ content of our measurements, the samples are graphitized by INSTAAR and then sent for analysis to the accelerator mass spectrometer at the University of California at Irvine. Results are reported in $\Delta^{14}C$ against the NBS Oxalic Acid I standard with an uncertainty of 1.8–2.5‰. Over the course of the GAUGE project a total of around 250 samples were analysed for $^{14}CO_2$. From this analysis we also received information about the stable isotopes $^{13}CO_2$, $CO^{18}O$, and $^{13}CH_4$, which

we do not report here. As part of the deployment of the Atmospheric Research Aircraft (described below) we collected glass flasks for the $^{14}CO_2$ and Tedlar bags for analysis of $^{13}CH_4$ by Royal Holloway, University of London. Using the aircraft allowed us to improve our knowledge of the spatial gradient of these gases. Samples were taken using an ORAC Metal bellows pump, fitted with



a pressure relief valve. For the glass flask sampling an adapter containing downstream pressure relief

valve was used to prevent the accidental over pressurizing of the glass flasks during flight sampling.

A preliminary study of $^{14}CO_2$ at Tacolneston during the GAUGE project has highlighted the benefits and difficulties associated with determining the fossil fuel content of $CO_2$ in the UK. The key outcome from the measurement program has suggested that the amount $CO_2$ originating from fossil fuel burning is not significantly different from model simulations using EDGAR emissions.

However, there were a number of difficulties associated with making these measurements. First, we used a number of assumptions and data corrections to account for terrestrial biosphere fluxes and nuclear emissions. For nuclear emissions, we expect that the applied correction can be significantly improved by provision of higher frequency emissions data from the nuclear industry. Second, the location of the sampling site, timing and frequency of measurements is paramount in determining

a strong enough $^{14}CO_2$ signal from fossil fuels to distinguish it from the background uncertainty. Many lessons were learnt in the GAUGE project that will allow for an improved and more robust sampling strategy to be applied to future measurements (Wenger et al, "Atmospheric radiocarbon measurements to quantify $CO_2$ emissions in the UK as part of the GAUGE project from 2014 to 2015" in preparation for *Atmos. Chem. Phys. Discuss.*, 2018).

**East Anglian Church Network**

A key objective of GAUGE was to improve understanding how to attribute observed variations of GHGs to particular sectors. To help address that objective we established a regional network of five sensors over East Anglia (Figure 1, Table 4) where there is a high density of crop agriculture, a sector with large seasonal emissions of $CH_4$ and $N_2O$ attributed to fertilizer application (Section

1). Developing this regional network supports the inference of higher resolution emission estimates (Manning et al., 2011). We used data from this network to determine how well we can distinguish between sources of $CH_4$ from spatially diffuse agricultural sources to point sources such as landfills.

We purposely distributed the network across East Anglia (Figure 1), comprising one atmospheric observatory (Weybourne) and three churches (Holy Trinity, Haddenham; All Saints, Tilney; and St

Nicholas, Glatton), and one wind turbine (Earl's Hall). East Anglia is one of several dense regions of UK agriculture. It was chosen for two reasons: 1) there is little variation in terrain height, simplifying boundary layer transport and mixing; and 2) all sites are within an hour of Cambridge, simplifying logistics associated with maintaining long-term sites. Additional criteria for site selection included sufficient sampling height (15–50 m for the East Anglia network, Table 4); remoteness from very

local sources of $CH_4$; easy accessibility for maintenance; and low running costs.

Figure 3 shows that the $CH_4$ mole fraction data collected from the three churches exhibit similar variations on diurnal, daily, and monthly timescales, suggesting that either the surrounding villages have similar sources and/or at least some of the observed variation reflect larger-scale variations. Observed variations of $CH_4$ at WAO are comparable to those at inland sites on seasonal timescales,



but are muted on faster timescales because it mainly observes clean upwind air. The shape of the
diurnal cycle at the church sites suggests that the boundary layer likely plays the dominant role.
Seasonal variations reflect changes in regional sources, boundary layer variations, and the OH sink.

Using the NAME-InTEM inverse model framework (Manning et al., 2011) we used the East
Anglian network to infer county-level $CH_4$ fluxes for Cambridgeshire, Norfolk, and Suffolk. Our

*a posteriori* fluxes were consistent with those from the UK National Atmospheric Emissions Inven-
tory (Connors et al, "Estimates of regional methane emissions from inversion modelling – a proof of
concept study," in preparation for *Atmos. Chem. Phys. Discuss.*, 2018). For this work it was difficult
to accurately estimate associated uncertainties because of difficulties associated with defining the
'background' $CH_4$ entering into the small, regional domain chosen. This difficulty will be avoided

when these data are included in larger, regional-scale inversions. We find that regional networks,
embedded within a nationwide network, show great potential for revealing additional spatial and
temporal details of emissions such as point source emissions from landfills (Riddick et al., 2017).
Such a regional network would best serve a national-scale network over regions where *a priori*
emission uncertainties are largest.

## 2.2  Mobile GHG Measurement Platforms

We use mobile platforms to help integrate measurements that are sensitive to different spatial scales.
The two principal platforms we use are the Rosyth-Zeebrugge North Sea ferry and the BAe-146
Atmospheric Research Aircraft. We also describe the deployment of balloon-borne sensors and a
fixed-wing unmanned aerial vehicle (UAV), as examples of GAUGE fostering new atmospheric

GHG measurement technology.

**North Sea Ferry**

We installed an eight-foot air-conditioned sea container on the Rosyth (56.02262°N, 3.43913°W)
to Zeebrugge (51.35454°N, 3.175863°E) ferry operated by DFDS Seaways. The container includes
a Picarro 1301 CRDS to measure mole fractions of $CH_4$, $CO_2$ and $H_2O$. This ship of opportunity

completes three return journeys per week traversing the North Sea at different times of day, thereby
minimizing temporal measurement bias that can sometimes complicate the analysis of data from
mobile platforms. The prevailing winds over the North Sea are westerly and southwesterly so that
measurements frequently sample the outflow from the UK, and also allow us to distinguish between
UK and mainland European emissions.

Figure 4 shows the view from the mobile laboratory, with sample inlets away from local sources on
the ferry. The initial installation was on 25th February 2014 on DFDS Seaways Longstone (now the
Finnmerchant) and ran until 15th April 2014. A weather station (Vaisala WXT 520) located on the
top deck provides basic meteorological data (air temperature, pressure, wind speed and direction);





geo-location information (latitude, longitude, ship speed, course) is obtained from a Garmin GPS
unit fixed to the roof of the sea container.

Figure 5 shows example $CH_4$ data for sailings in March, April, July, and September 2014, which
shows a dynamic range that reflects geographical variations in sources. Differences between sailing
reflect changes in seasonal emissions and prevailing meteorology. Figure 5 shows instances when
observed values are influenced by emissions from the UK and the North Atlantic background during
spring and summer (Figure 5a,b), and when observed values are influenced by high emissions from
Germany and central Europe (Figure 5c) and by lower emissions from Scandinavia (Figure 5d). A
more detailed description of the instruments and the data interpretation can be found in C. Helfter et
al, "Temporal variability in country-scale greenhouse gas budgets using a mass balance approach,"
in preparation for *Atmos. Chem. Phys. Discuss.*, 2018.

**BAe-146 Atmospheric Research Aircraft**

We use the NERC/Met Office Atmospheric Research Aircraft (ARA), operated by AirTask Group
Ltd, to provide vertical profile distributions of atmospheric GHGs over and around the British Isles.
The specific objectives of deploying the ARA include: 1) collect a snapshot of precise and traceable
GHG concentration distributions over and around the UK; 2) integrate atmospheric GHG informa-
tion collected by tall towers, ferry transects, and space-borne instruments; 3) define and execute
sampling experiments to enable measurement-led quantification of GHG fluxes at the regional scale
($\mathcal{O}$(100 km)); and 4) define and execute sampling experiments to challenge Earth system models
and flux inversion models in terms of better understanding model atmospheric transport error and
surface emission distribution.

The ARA is a BAe-146-301 aircraft that has been converted to a mobile laboratory, including a
variety of forward and backward facing external inlets so that air can be sampled by instruments
within the main cabin. It also includes a number of ports that can host remote sensing instruments.
Table 5 describes the instruments that we deployed during GAUGE, including in particular instru-
ments that measure $CO_2$, $CH_4$ and $N_2O$, and a small complementary suite of other trace gases and
thermodynamic parameters. We made continuous measurements of $CO_2$ and $CH_4$ at a frequency of
1 Hz using a Fast Greenhouse Gas Analyser (FGGA, Los Gatos USA). For a detailed description of
the FGGA, including its operating principles, data processing and calibration, we refer the reader to
O'Shea et al. (2013). We also collect 1 Hz measurements of $N_2O$ and $CH_4$ from a quantum cascade
laser absorption spectrometer (Aerodyne Research Inc., USA). Further details of the instrument are
described by Pitt et al. (2016). We use the Met Office Airborne Research Interferometer Evaluation
System (ARIES), a Fourier transform infrared spectrometer, to retrieve partial columns of $CH_4$ and
$CO_2$ and vertical profiles of $H_2O$ and temperature. Further details about ARIES can be found in
Allen et al. (2014). Other instruments listed in Table 5 are core ARA science instruments, which are
described in Allen et al. (2011) and references therein.



During GAUGE we conducted a total of 16 individual flight sorties over/around mainland UK and Ireland between May 2014 and March 2016, comprising over 65 hours of atmospheric sampling. These flights are summarized in Table 6 and Figure 6. A typical flight sortie coordinated upwind and downwind sampling of a target flux region (e.g., the London metropolitan area), based on the prevailing boundary layer wind direction, to attempt sampling of airmasses that have been im-
pacted by regions with GHG emissions and uptake. We also designed flights to sample outflow from mainland UK and continental Europe, and outflow from the Irish and North seas on days with strong westerly flow regimes, e.g. J. Pitt et al, "Development of a method to assess $CH_4$ flux using aircraft and ground-based sampling: a case study for the British Isles on 12 May 2015," in preparation for *Atmos. Chem. Phys. Discuss.*, 2018.

To capture regional emissions during GAUGE, we collected measurements that were mostly in the boundary layer, as defined by in-flight thermodynamic profiling, which was typically below 2 km altitude. Occasionally, to characterize long-range transport of pollutants into our study region, we collected measurements during deeper vertical profiles into the free and upper troposphere. Other flight profiles included surveys around Britain and Ireland and flying around tall towers, as described
below.

Figure 6 shows a summary plot of the $CO_2$ and $CH_4$ data collected during GAUGE. In particular, it illustrates the horizontal and vertical spatial coverage of the aircraft sampling, and the dynamic range of mole fractions sampled. These observed variations are due to differences in flight altitude and the time of year of the superimposed flights (Table 6), differences in airmass history, and the
spatial and temporal variability of local and regional fluxes across seasons and sources.

Table 7 shows a comparison between aircraft and tall tower data during aircraft fly-pasts. The mean difference between $CO_2$ ($CH_4$) observations at all tall tower sites measured during 12 individual flights is 0.72±1.69 ppm (-1.22±12.54 ppb). Taking into account that the majority of the flights took place during summer months, the magnitude of these difference is as expected with generally
lower $CO_2$ and higher $CH_4$ mole fractions closer to the ground and more sensitive to local fluxes.

**Balloon $CO_2$ Sondes**

Balloons offer an alternative platform for the collection of vertical profiles of GHGs, building on the approaches used widely by the meteorological and stratospheric communities. Here, we describe some of the first balloon launches of small-scale $CO_2$ sensor technology that have been adapted for
atmospheric sciences. ChemSonde is a balloon-based instrument, developed as part of a collaboration between the University of Cambridge, SenseAir (Sweden), with additional input from Vaisala (Finland) and Alphasense (UK). The aim of ChemSonde is to provide a cheap method for measuring $CO_2$ concentrations from the surface to $\simeq 30$ km on a global scale by using the existing radiosonde infrastructure, and to form the basis of a calibration/validation programme to support space-borne
observations of GHGs.



The instrument consists of a small, sensitive nondispersive infrared $CO_2$ sensor developed by SenseAir, Sweden, (www.senseair.se) which has been adapted for atmospheric measurements. The instrument sampling is 1 Hz with data transmitted to the Vaisala MW41 ground station via the radiosonde. The corresponding vertical resolution of the collected data is 4–5 m. The dimensions and weight of the instrument package are approximately $150 \times 150 \times 300$ mm and 1 kg, respectively. Heavy-duty cable ties are used to seal the enclosure and secure the radiosonde to the outside. A 1200 g balloon (TOTEX, Japan) is used for lifting the payload.

Figure 7 shows preliminary data from two ChemSonde launches from WAO on the 14th April 2016 to test the viability of the system. Met Office surface analysis charts (not shown) indicate that the UK was under the influence of a low pressure anticyclone in the North-Atlantic, transporting moist air over the southern half of the UK, during the period of measurements. A low-level stratus cloud deck, with drizzle, and low SW winds predominated over WAO during the morning of the 14th April, with light winds and steady rain during the afternoon. The first instrument was launched at 1039 UTC, and the second at 1430 UTC. For brevity, we only show data to 10 km. The sharp decrease in $CO_2$ from near-surface altitudes to $\simeq 1$ km during the morning launch, and the increase in boundary-layer $CO_2$ concentrations from morning to afternoon launches suggest some local influence. We also noticed that some small-scale increases in $CO_2$ (1.8 km and 7.5 km from the morning launch and 2.5 km from the afternoon launch) correspond to increased relativity humidity, indicating possible cloud layers. NOAA HYSPLIT 48-hour back trajectories (Stein et al., 2015) initialized at these lower and mid troposphere altitudes (not shown) indicate that we are sampling background maritime air over the North Atlantic that has been lofted prior to interaction with land surfaces. Differences in relative humidity close to 6 km suggest that the morning cloud structure has been dissipated by the stronger afternoon winds. We attribute the 4–5 ppm difference between $CO_2$ instruments above 6.5 km to problems with the zero baseline drift, and to a faulty span measurement during the afternoon pre-launch preparation. Further studies with ChemSonde are planned, with emphasis on improving design, operation and the post-processing of data.

**Unmanned Aerial Vehicles for Hotspot Measurement Campaign**

UAVs represent a new atmospheric measurement platform for studying atmospheric GHGs. They can be deployed rapidly to provide vertical information across a horizonal dimension $\mathcal{O}(100$ m$)$. Within GAUGE, researchers used a variety of measurement technologies, including fixed-wing and rotary UAVs, to develop and refine new methods to use atmospheric measurements to quantify $CH_4$ and $CO_2$ emission from a landfill site (Riddick et al., 2016; Sonderfeld et al., 2017; Allen et al., 2017; Riddick et al., 2017). This represents one of the first demonstrations of using UAVs to sample GHG emissions. The reader is referred to Allen (2014); Allen et al. (2015) for further details of the underlying technology.



We conducted a two-week measurement campaign at a landfill site near Ipswich, England (operated by Viridor Ltd) in August 2014. This campaign brought together researchers from Universities of Bristol, Cambridge, Denmark Technical University, Edinburgh, Leicester, Manchester, Royal Holloway University of London, Southampton, and Ground Gas Solutions (GGS) Ltd. The landfill

includes historic, capped and active, open landfill cells, a leachate plant, a gas collection network and gas burning energy generation facility.

We equipped the site with a 20 m eddy covariance flux tower, three Los Gatos Research ultra-portable greenhouse gas ($CO_2$ and $CH_4$) analysers (triangulated across the capped and open cell areas), a closed path FTIR, and five 3-D sonic anemometers to characterize flow over the site. Con-

ventional walkover flux surveys were conducted by GGS and dynamic automated flux chambers were operated on the flanks of the capped landfill area to investigate seeps under the capped area where this met an active cell. Tracer releases of perfluorocarbon and acetylene were also conducted from various key points across the site to allow proxy flux calculations from mobile (public road) plume sampling downwind. Specific experiments and instrument-siting were designed on each day

of the intensive period in response to weather (especially wind) conditions to characterise inflow and outflow from different areas of the site. We deployed a fixed-wing UAV equipped with a $CO_2$ sensor around the site. We also launched a tethered rotary UAV, which sampled air up to 120 m above the local terrain and analyzed using ground-based instruments via a 150 m length of Teflon tube. This configuration allowed us to sample vertical profiles of $CH_4$ and $CO_2$ over the landfill site.

We also established a fixed-site monitoring station measuring $CO_2$ and $CH_4$ mole fractions to put the campaign into a longer temporal context, to help test plume inversion techniques, and to test the efficacy of continuous *in situ* monitoring to generate flux climatologies (Riddick et al., 2016, 2017). Sonderfeld et al. (2017) demonstrate how to combine computational fluid dynamics model (which accounts for topographical data from a 3-D LiDAR survey data) with continuous *in situ* FTIR

measurements to infer and apportion fluxes across the surface area of the landfill site. They showed in particular the ability of this approach to distinguish between individual emission regions within a landfill site, allowing better source apportionment compared with other methods that derive bulk emissions.

Our UAV deployment during this experiment has since led to further refinements to the method

and platform, and to our use of similar technology to infer fluxes from other UK landfills (Allen et al., 2017). A recent validation of a new mass balancing algorithm based on UAV sampling of a known $CH_4$ release rate demonstrated that a 20-minute flight on a single rotary UAV flight can reproduce the known release rate with an mean accuracy of 14% and an ($1\sigma$) uncertainty of <40% (Shah et al., 2017). Collectively, these measurements allowed us to test and compare a wide range of

established and novel sampling technologies and flux quantification approaches. It also allowed us to examine how to optimize different combinations of data to determine net bulk (whole-site) GHG fluxes.



### 2.3 Space-borne Observations of GHGs

Satellites provide global, near-continuous and multi-year measurements of GHGs that are used to
infer GHG fluxes on sub-continental scales, and to provide boundary conditions for regional at-
mospheric transport models. Within GAUGE, we explore the potential of short-wave IR (SWIR)
column measurements of $CO_2$ and $CH_4$ from the Japanese Greenhouse Gases Observing SATellite
(GOSAT) and thermal IR column measurements of $CH_4$ from the European Infrared Atmospheric
Sounding Interferometer (IASI). For the sake of brevity, we describe here only the pertinent details
of GOSAT and IASI and refer the reader to other studies dedicated to these satellite instruments (e.g.
Kuze et al. (2009); Clerbaux et al. (2009)).

GOSAT is the first space-borne mission dedicated to measuring GHGs. It was launched in a sun-
synchronous orbit with a local overpass time of 1300 by the Japanese Space Agency (JAXA) in
January 2009 (Kuze et al., 2009). We use the Thermal And Near-infrared Sensor for carbon Observa-
tion (TANSO) FTS that observes atmospheric spectra and the Cloud and Aerosol Imager (CAI) that
provides multi-spectral imagery and coincident cloud and aerosol information (Kuze et al., 2009).
TANSO-FTS has a ground footprint of approximately 10.5 $km^2$ and returns to the same point every
three days. For illustration, we show GOSAT SWIR dry-air column-averaged $CH_4$ mole fractions
that are inferred from version 7.0 of the proxy retrieval developed by the University of Leicester
(section 3). These data are sensitive to changes in atmospheric $CH_4$ in the lower troposphere. The
proxy retrieval method simultaneously fits $CH_4$ and $CO_2$ spectral features in nearby wavelengths.
The underlying idea is that taking the ratio of the $CH_4$ and $CO_2$ fitted in nearby wavelength regions
effectively removes spectral artefacts common to both $CH_4$ and $CO_2$ (e.g., scattering). The conven-
tional method of using these data is to multiply the ratio by model $CO_2$, assuming that $CO_2$ varies in
space and time less than $CH_4$. The resulting proxy $XCH_4$ data have been evaluated extensively using
data from the Total Carbon Observing Network (Parker et al., 2011, 2015).

IASI is one of a series of Fourier Transform Spectrometer (FTS) instruments on the polar-orbiting
meteorological MetOp platforms (Hilton et al, 2012) designed primarily for operational meteorol-
ogy. There are two IASI instruments currently operating: MetOp-A was launched on 19th October
2006 and MetOp-B was launched on 17th September 2012. IASI has an across-track measurement
swath of 2,200 km, resulting in near-global coverage twice a day with a local solar overpass time
of 0930 and 2130. It measures three spectral bands that span a range of thermal IR wavelengths
from 4 microns to 15.5 microns (Clerbaux et al., 2009), which are most sensitive to $CH_4$ in the mid-
troposphere. Vertical profile retrievals of column-averaged volume mixing ratios of atmospheric $CH_4$
have been inferred using optimal estimation from IASI spectra by the Rutherford Appleton Labora-
tory (Siddans et al., 2017). The retrieval produces two pieces of information in the mid/upper tropo-
sphere each with a single retrieval precision of 20–40 ppbv. Differences between IASI and GOSAT
$CH_4$ are within 10 ppbv except over southern mid-latitudes where IASI is lower than GOSAT by
20–40 ppbv (Siddans et al., 2017).





The spatial coverage of satellite SWIR observations of $CO_2$ and $CH_4$ over the UK is limited
mainly by cloud-free scenes that are themselves determined by the spatial resolution of the instru-
ments and the repeat frequency of the orbits. Currently, there are insufficient cloud-free data to
overtake the information provided by the *in situ* measurements. However, we will soon have daily
$CH_4$ measurements from TROPOMI aboard Sentinel-5P, launched 16th October 2017. Data from
future and planned missions represent at least an order of magnitude more satellite data than we
have now. Until then, these data GOSAT represents constraints on larger-scale sub-continental $CO_2$
and $CH_4$ flux estimates (e.g. Feng et al. (2017)).

### 2.4  Intercalibration activities

Linking measurements in the GAUGE network to a common calibration scale ensures compara-
bility of these measurements, and simultaneously linking them to a common set of traceable gas
standards ensures they are also compatible with ongoing international GHG measurement activities.
Two prominent examples of such activities include the pan-European Integrated Carbon Observing
System (ICOS, https://www.icos-ri.eu/) and the Integrated Global Greenhouse Gas Information Sys-
tem (IG[3]IS, https://goo.gl/4t1x6i). The GAUGE project encompassed a large number of data streams
collected using a range of instrumental techniques and at a variety of temporal resolutions, increas-
ing the risk of compatibility and comparability errors. Inversion methods used in GAUGE to infer
GHG fluxes from atmospheric mole fraction measurements are particularly sensitive to site biases
and offsets (Law et al., 2008). Consequently, ensuring comparability and assessing compatibility
was key to the success of GAUGE.

As far as possible we ensured measurement comparability by linking all observations directly to
common WMO calibration scales, but due to the historical nature of some data records this was not
uniformly possible. All $CO_2$ measurements collected within the project were linked to the WMO
x2007 scale. All $CH_4$ measurements, other than MHD GC-FID (Table 2) that uses the Tohoku scale,
were calibrated to the WMO x2004A scale. In contrast, $N_2O$ measurements used either the SIO-98
scale (MHD and the rural tall tower sites BSD, HFD, RGL, TAC and TTA) or the WMO x2006A
scale (all other locations).

### 3  Numerical Models of Atmospheric GHGs

Figure 8 shows the modelling strategy we employed to quantify the magnitude, distribution and un-
certainty of UK emissions of GHGs from different sectors. We use models of atmospheric chemistry
and transport, using prescribed *a priori* flux estimates, to describe the relationship between sector
emissions of GHGs and atmospheric variations observed by the fixed and mobile GHG measurement
platforms used during GAUGE (Figure 1). These models, which account for instrument-specific
sampling, constitute the forward model. Inverse models infer the magnitude and uncertainty of re-



gional flux estimates by fitting the forward model to observations, accounting for their respective
uncertainties.

Because of the complex physical and chemical relationships between the surface fluxes and the atmospheric observations, and because of the assumptions embedded within individual models, we use a range of atmospheric transport models and inverse methods to mitigate criticism that our results depend only one model.

## 3.1 Atmospheric Chemistry Transport Models

Table 9 summarizes the three different chemical transport models (CTMs) and one atmospheric dispersion model that we use to interpret the GAUGE data. All models are well established and have been used to interpret a wide range of atmospheric GHG measurements.

**Brief Description of Individual Models**

We use the following models: 1) the Goddard Earth Observing System atmospheric Chemistry transport model (GEOS-Chem) (Feng et al., 2011; Fraser et al., 2013; Deng et al., 2014; Feng et al., 2017); 2) the Model for OZone and Related chemical Tracers (MOZART) (Emmons et al., 2010); 3) the TOMCAT model (Wilson et al., 2016; McNorton et al., 2016; Monks et al., 2017); and 4) the Numerical Atmospheric dispersion Modelling Environment (NAME) (Jones et al., 2007). These models
vary in their basic methodologies for representing atmospheric transport, parameterisations of physical atmospheric processes, and in their horizontal and vertical resolutions. We have ensured, as much as possible, that we use common model boundary conditions (e.g., flux inventories and lateral boundary conditions for regional models). Model differences therefore provide us an opportunity to quantify the impact of model error on describing observations and consequently on inferred GHG
flux estimates. For further details about an individual model, the reader is encouraged to consult the model-specific literature as provided above.

For the purpose of this overview of GAUGE and as part of our model assessment within GAUGE, we ran global 3-D experiments to describe observed variations of $CO_2$, $CH_4$ and $N_2O$ from 2004 to 2016, including the main GAUGE measurement period of 2014–2015, inclusively. The CTMs used
common flux estimates and chemical loss fields as described below. Preparation of these estimates, collected from different sources, were regridded to the different model resolutions (Table 9), ensuring that the total emitted mass was conserved. The CTMs also used common atmospheric mole fraction initial conditions for 2003.

To describe anthropogenic emissions of $CO_2$ from 2003 to 2009, we use the Carbon Dioxide Infor-
mation Analysis Center (CDIAC) inventory (available online at http://cdiac.ornl.gov/trends/emis/overview.html). In later years, we repeat values from 2009. We use the NASA-CASA biosphere model (Olsen and Randerson, 2004) to describe terrestrial biospheric fluxes, 2003–2015, including biomass burning





emissions. Climatological ocean fluxes of $CO_2$ are taken from Takahashi et al. (2009), covering the period 2003–2011.

The formulation of our $CH_4$ simulations generally follows Wilson et al. (2016); McNorton et al. (2016). We use updated anthropogenic $CH_4$ emissions from the Emission Database for Global Atmospheric Research (EDGAR) v4.2FT inventory Olivier et al. (2012) , covering the period 2000–2010. We repeat 2010 emissions for years beyond 2010. Biomass burning emissions were taken from the Global Fire Emissions Database (GFED) v3.1 inventory (van der Werf et al., 2010). Wetland and

rice emissions were taken from Bloom et al. (2012). Other natural emissions, including the soil sink (treated as a negative flux) were taken from the TransCom $CH_4$ model intercomparison (Patra et al., 2011). We use monthly 3-D mean OH fields taken from Patra et al. (2011) to describe the main atmospheric sink of $CH_4$. Reaction rates are taken from Sander et al. (2006). Stratospheric loss of $CH_4$ due to reaction with $O(^1D)$ and Cl radicals are based on loss rates taken from the Cambridge 2-D

model (Velders, 1995). The resulting atmospheric lifetime of $CH_4$ is $\simeq$10 years, which is determined mainly by the tropospheric OH sink.

        Fluxes for our $N_2O$ simulations are taken from four broadly defined source categories: natural soils (Saikawa et al., 2014), agricultural and other anthropogenic emissions (Olivier et al., 2012), ocean fluxes (Manizza et al., 2012), and biomass burning (van der Werf et al., 2010). We parame-

terized an offline stratospheric loss of $N_2O$ in each model using photolysis and $O(^1D)$ climatologies (Thompson et al., 2014). We did not consider this sink for NAME because of the short duration of model runs compared to the atmospheric lifetime of $N_2O$ ($\simeq$120 years). The relatively long atmospheric lifetime of $N_2O$, determined by stratospheric sinks, means that interpreting observed tropospheric variations of $N_2O$ presents different challenges to interpreting observed variations of

$CH_4$.

**Assessment of Model Performance using Large-scale Independent data**

To assess the global-scale GAUGE models we use data that are representative of large spatial and temporal scales. In particular, we use surface mole fraction data from NOAA/ESRL and column data from the GOSAT and IASI satellite instruments (Section 2). We use these data to evaluate the

three free-running CTMs, described above, by sampling each model at the time and location of each observation.

        Figure 9 shows that the models reproduce the broad scale zonal-mean distribution of $CO_2$ and $CH_4$. Given the common set of source and sink terms, model divergence will mostly reflect differences in atmospheric transport. Generally, the largest model biases for $CO_2$ are at mid/high northern

latitudes where the emissions are largest. Model divergence is highest at these latitudes during northern winter months, with GEOS-Chem having the largest model bias during these months. Model performance generally improves in the northern summer months with model differences typically within a few ppm and much closer to the observations. The model spread supports our strategy of



using different models to infer GHG fluxes. For CH$_4$, the models have a similar level of skill. None

of the models reproduce the observed inter-hemispheric gradients, likely due to errors in the *a priori*
distribution of emissions used by the inventories. The model spread is largest in January with a value
of 45 ppb. Model performance for N$_2$O is the most variable, although this partly reflects that N$_2$O has
the smallest observed inter-hemispheric gradients of the three gases. The maximum model range is
1.4 ppb and 1.7 ppb in January and July, respectively. The GEOS-Chem and MOZART models have

gradients similarly small in the southern hemisphere and tropics, while TOMCAT is much larger.

Figure 10 shows that MOZART and GEOS-Chem have similar vertical distributions of CH$_4$ dur-
ing July, displaying a stronger vertical gradient from the surface to 400 hPa than the TOMCAT
model. This corresponds to higher northern hemispheric mole fraction values. During July, the three
models all display different rates of vertical transport throughout the northern hemisphere tropo-

sphere. TOMCAT has a slight gradient between the surface and 600 hPa, and a much steeper gradient
above; MOZART displays the opposite behaviour; and GEOS-Chem lies between those extremes.
Differences in atmospheric transport are important and for some gases can represent a substantial
fraction of the signal. Our use of multiple models and combining the resulting analysis improves our
ability to quantify the uncertainty of our results.

We also evaluate the models using the GOSAT Proxy XCH$_4$ V7.0 data product developed by
the University of Leicester (http://www.esa-ghg-cci.org/) and the IASI MetOp-A thermal IR V1.0
XCH$_4$ data products developed by the Rutherford Appleton Laboratory (http://dx.doi.org/10.5285/
B6A84C73-89F3-48EC-AEE3-592FEF634E9B).

Figure 11 shows the spatial coverage provided by both instruments during June–August 2014.

The sparser coverage of GOSAT observations reflects its sensitivity to clouds and aerosols. Mea-
surements over the ocean used a glint observing model that takes advantage of specular reflection
and its associated high signal to noise ratio. Despite GOSAT and IASI observing different parts of
the atmosphere there are many common features associated with fossil fuel extraction/combustion
(North America, China, and parts of Saudi Arabia), wetlands (South America, Africa, and part of

India and China), and rice paddies (mostly India and China). Both GEOS-Chem and TOMCAT
model reproduce the broad spatial distributions of GOSAT and IASI CH$_4$ observations (not shown),
with negative global mean model biases that are approximately 10 ppb for GOSAT and between
1 ppb (GEOS-Chem) and 10 ppb (TOMCAT) for IASI. These biases mainly reflect errors in *a priori*
surface emissions, but also errors in modelling stratospheric CH$_4$ (e.g. Alexe et al. (2015)).

## 600 3.2 Inverse Methods

The ultimate objective of GAUGE is to characterize the magnitude, distribution, and uncertainty of
UK GHG emissions. Relating *a priori* GHG flux estimates to the atmosphere sampled at the time and
location of observations is called the forward problem (Figure 8). The corresponding inverse problem
refers to the process of relating observed atmospheric measurements to the underlying geographical





distribution of GHG fluxes. Each of the atmospheric transport models listed above employ their own inverse method, as described below.

Inferring $CO_2$, $CH_4$, and $N_2O$ fluxes directly from atmospheric observations is generally an ill-posed inverse problem, with a wide range of scenarios that could fit these data. *A priori* information is used to regularize the problem (Figure 8).

The results of inverse modelling are typically dependent on the distribution of the observations used. For example, the sparsity of data at low latitudes places a limit on our ability to infer GHG fluxes over geographical regions that are not well sampled, e.g. tropical ecosystems. The spatial and temporal density of GHG measurements collected during GAUGE allows us to constrain *a posteriori* emission estimates on devolved UK administration scale and on sub-annual timescales.

Although Bayes' theorem provides the basis for each of the inverse modelling techniques used in GAUGE, each approach employs a slightly different methodology to infer optimized surface fluxes. As we have already seen there can be relatively large differences in atmospheric transport models. Indeed, the errors associated with atmospheric transport models are typically the largest source of error in estimating GHG fluxes.

In the interest of brevity, we only briefly introduce the inverse methods employed within GAUGE and refer the reader to dedicated cited papers on the techniques.

The global and nested GEOS-Chem model is linked with an ensemble Kalman filter (Feng et al., 2009, 2011, 2017). This approach does not require that we linearize the model but assumes approximate Gaussian statistics. The ensemble Kalman filter approach allows us to include easily estimates of model atmospheric transport error. Flux estimates are resolved on geographical regions informed by the ability of the data to independently estimate fluxes on those spatial scales. Over the UK, fluxes are estimated on pre-defined aggregated county levels and on a weekly scale. Weekly values are subsequently aggregated to longer timescales to minimize autocorrelation between successive flux estimates.

The inverse version of the TOMCAT model, INVICAT (Wilson et al., 2014) uses a variational inversion method based on 4D-Var. This approaches uses the adjoint version of the forward model to minimize the *a posteriori* fit between the model and data. This is an iterative method that can sometimes require a large number of iterations before convergence. Consequently, we resolve *a posteriori* emissions using TOMCAT at a spatial resolution of $2.8°$.

The NAME model uses the InTEM inverse method, building on Manning et al. (2011) but now posed in a hierarchical Bayesian method in which the basis function decomposition of the flux space, and the model and *a priori* uncertainties, are explored using reversible-jump MCMC (Ganesan et al., 2014; Lunt et al., 2016). For the MOZART model we used a hierarchical Bayesian method based on Ganesan et al. (2014). InTEM estimates emissions across a north west European domain at horizontal resolutions from 25 km to 100s km, depending on the frequency of sampling different regions. Boundary conditions are solved within each NAME inversion, following Ganesan et al. (2015) for



InTEM and Lunt et al. (2016) for the MCMC approach. Monthly UK emission estimates of $CH_4$ and $N_2O$ were estimated for the period 2013–2016 and compared to the reported inventory.

Our GAUGE inverse model studies generally include a series of factorial experiments that allowed us to explore the relative importance of individual and collective data to estimate UK $CO_2$ and $CH_4$ flux estimates. Based on these experiments we define a control experiment. We test the robustness of our results by comparing results from using half/double assumed measurements uncertainties. UK *a posteriori* flux estimates for $CO_2$ and $CH_4$ are currently being prepared for publication: Lunt et al, "Evaluating national methane emissions using atmospheric observations," in preparation for *Atmos. Chem. Phys. Discuss.*, 2018. and Palmer et al, "Using atmospheric measurements to verify UK net fluxes of carbon dioxide," in preparation for *Atmos. Chem. Phys. Discuss*, 2018. Broadly speaking, we have estimated net $CO_2$ fluxes using regional and global-scales, but have been unable to attribute those fluxes to specific sectors; for $CH_4$, using the continental-scale data and the regional network data, we have begun to improve our understanding of sector emissions; and for $N_2O$, which has the small atmospheric gradients due to its long atmospheric lifetime, we have not begun to analyze the data collected within GAUGE.

## 4 Concluding Remarks

The main objective of the Greenhouse gAs Uk and Global Emissions (GAUGE) project was to estimate the magnitude, distribution, and uncertainty of UK emissions of three atmospheric greenhouse gases (GHGs): carbon dioxide ($CO_2$), methane ($CH_4$), and nitrous oxide ($N_2O$). To achieve that objective, we established an inter-linked measurement and data analysis programme of activities from 2013 to 2015. These activities substantially expanded on existing measurements and data analysis. Some measurements that were established as part of GAUGE have continued beyond 2015. The primary motivation for GAUGE was to develop a measurement-led system to verify UK GHG emissions in accordance with the UK Climate Change Act 2008. GAUGE also lays the foundations for estimating nationally determined contributions as part of the Paris Agreement.

Emissions of $CO_2$, $CH_4$, and $N_2O$ represented 97% of UK GHG emissions during 2015 (the latest budget estimates available from the UK government). These emissions originate from a variety of sectors, including energy supply, transport, business, residential, agriculture, waste management, and other. These emissions are very different in nature, ranging from point sources to large-scale, diffuse sources. We considered this heterogeneity of course when we designed the GAUGE measurement programme.

The backbone of GAUGE is a network of measurements that are collected at height from telecommunication masts, tall towers, distributed across the UK. These measurements are typically collected at multiple inlet heights (100–300 m) above the local terrain (and sources) so they have a reasonable fetch suitable for quantifying sub-national scale GHG fluxes. GAUGE added two tall tower sites



to the UK Deriving Emissions linked to Climate Change (DECC) tall tower network. The DECC
network was established in 2012 to estimate GHG emissions from the UK devolved administrations.
The GAUGE sites included a site on the North Yorkshire Moors, with sensitivity to the Greater
Manchester-Leeds-Liverpool-Sheffield region, and in East Sussex that has sensitivity to emissions
from London.

We collected data on a commercial ferry that travelled regularly between Rosyth, Scotland, and
Zeebrugge, Belgium. This mobile measurement platform provided information on UK and mainland
European outflow of GHGs, which complemented the tall tower data. Using a regional tower net-
work over East Anglia, comprising mostly of measurements collected on Church steeples, we found
additional spatial and temporal flux distributions over the region could be achieved. We chose East
Anglia because it is where there is a high density of agriculture, and where the local terrain is rela-
tively flat so that church steeples often represent the highest local landmarks. As part of GAUGE we
deployed the UK Atmospheric Research Aircraft for a limited number of flights around and across
the UK. These data have been used to study the transport of atmospheric GHGs on local to regional
spatial scales.

To explore how the UK GHG measurement network could develop in the future, we incorporated
new technologies and new measurement platforms into the GAUGE programme. We deployed small
sensors that were launched on a small number of sonde launches, which offer a potentially new
way to obtain vertical distributions of GHGs. We also used unmanned aerial vehicles as part of a
larger measurement campaign to characterize GHG emissions from a landfill, helping to pave the
way for using this technology more generally within larger-scale GHG emission experiments. We
also explored how we can use satellites effectively to estimate UK GHG fluxes. The spatial and
temporal coverage of clear-sky measurements over the UK from current SWIR instruments, which
are sensitive to changes $CO_2$ and $CH_4$, are too sparse to provide competitive constraints on $CO_2$
fluxes. We anticipate this situation will slowly change with new instruments (e.g. TROPOMI) and
proposed mission concepts (e.g. Copernicus $CO_2$ service) that will result in higher spatial resolution
and consequently more cloud-free scenes.

We used a range of global and regional atmospheric transport models linked with inverse methods
to interpret the atmospheric GHG observations. We showed that these models have skill in reproduc-
ing observed atmospheric $CO_2$ and $CH_4$ variations on hemispheric scales, but disagree with $N_2O$
observation due to much small gradients that reflect its longer atmospheric lifetime. This multi-
model approach was adopted to help study the model spread in *a posteriori* GHG fluxes, and to
study the relative importance of individual data to estimate UK GHG fluxes. For this work, we refer
the reader to the dedicated papers.

We approached source attribution in two ways. First, we used the regional-scale network to im-
prove the distribution of $CH_4$ fluxes due to agriculture, taking advantage of reasonable spatial disag-
gregation of this source over East Anglia. We also established an isotope measurement programme,



including concurrent measurements collected at Mace Head, Ireland, and Tacolneston, East Anglia. Data from these two sites provided a crude meridional gradient over the UK. Our sampling approach was designed, using the prevailing wind direction over the UK, to determine the gradient due to fossil fuel $CO_2$. Despite our best efforts, neither approach to source attribution was definitive. For example, our analysis of radiocarbon was compromised by the influence of the nuclear power sector. We anticipate the development a more optimal sampling approach is possible by working more closely with this sector to avoid instances when sampled air masses are dominated by upwind the nuclear source.

GAUGE represents a first concerted attempt by the UK science community to quantify nationwide GHG fluxes. We have laid the foundations of measurement infrastructure that moves forward with a better understanding of the advantages and disadvantages of individual GHG data. The post-GAUGE tall tower network has continued. For instance, the UK DECC network has adopted North Yorkshire site, which provides valuable flux information about northern England and to a lesser extent southern Scotland, and the National Physical Laboratory now runs the tall tower at Heathfield. We also anticipate a growing role for satellite observations, which are free at the point of delivery, as new instruments provide better spatial coverage and probabilistically a higher number of cloud-free scenes. Data analysis will continue as improved models and inverse methods progressively better describe the physical and chemical processes that determined atmospheric GHGs. The UK is a geographically small country and plays a proportional role in the Paris Agreement, but we expect the design of GAUGE can be scaled upwards to larger geographical regions, taking advantage of specific technologies relevant to the sectors that dominate continental GHG budgets.

*Acknowledgements.* The GAUGE project was funded by the Natural Environment Research Council under grant reference NE/K002449/1. We gratefully acknowledge the cooperation and efforts of the station operators Gerard Spain and Duncan Brown at Mace Head monitoring station, and Stephen Humphreys at the Tacolnestion tall tower station. We also thank the Physics Department, National University of Ireland, Galway, for making available the research facilities at Mace Head. We thank the Parish councils of Holy Trinity, Haddenham, Cambridgeshire; All Saints, Tilney St Lawrence, Norfolk; and St Nicholas, Glatton, Cambridgeshire as well as Greencoat Capital for their kindness and assistance in hosting instruments in our East Anglian network. The Diocese of Ely was instrumental in facilitating their involvement, and we especially acknowledge the assistance of Bill Murrells, David Ogilvie, and the Revs Fiona Brampton, Nigel Cooper, Martin Dale, Barbara Pearman and Rosie Ward. We thank collaborators at the Universities of Southampton, Royal Holloway University London, and Ground Gas Solutions Ltd for their support of the landfill case study, and Viridor Ltd for providing on-site support and facilitated access to their operational site; staff (including pilots) at the Facility for Airborne Atmospheric Measurement, Airtask Ltd, and Avalon Engineering Ltd for their support in conducting airborne fieldwork; DFDS Seaways for authorising the research activities on board the Rosyth-Zeebrugge commercial ferry; Captains and crews of the Longstone (now the Finnmerchant) and Finlandia Seaways for access to the ships and for supporting the day-to-day research operations; Ray Freshwater and Bin Ouyang (Chemistry



Department, University of Cambridge) for invaluable technical assistance and design of the ChemSonde motherboard, and for processing the ChemSonde raw $CO_2$ data; and Grant Forster (University of East Anglia) for access and assistance at NCAS-funded Weybourne Atmospheric Observatory. L. F. also acknowledges funding from the NERC National Centre for Earth Observation. P.I.P. gratefully acknowledges his Royal Society
Wolfson Research Merit Award. J.P. and S.C. were funded by NERC PhD studentships NE/L501/591/1 and NE/J500070/1, respectively. N.H. also received support from Defra and the Royal Society. The operation of all tall tower stations was funded by BEIS through contract GA01103. The ChemSonde work at Cambridge also funded under NERC grant number NERC NE/K005855/1.

**Appendix A: Tall Tower Site Descriptions**

Table 1 describes the basic characteristics of each site. The MHD atmospheric research station is situated on the west coast of Ireland. MHD receives well-mixed air masses from prevailing southwesterly winds across the North Atlantic (on average 37% of the time (Grant et al., 2010)), providing a good mid-latitude Northern Hemisphere background signal. The resulting timeseries provides an essential baseline for the combined UK GHG measurement network. The area immediately sur-
rounding MHD is generally wet, boggy with areas of exposed rock and is sparsely populated with very low associated anthropogenic emissions (Dimmer et al., 2001). The closest city to MHD is Galway, which lies 55 km east of MHD and has a population of 75,000.

RGL is a rural UK site located 30 km from the border of England and Wales. It is 16 km southeast of Hereford (population 55,800), and 30 km southwest of Worcester (population 98,800), in Here-
fordshire, UK (Office for National Statistics, 2012). The land surrounding the tower is primarily used for arable, livestock and mixed farming purposes (Department of the Environment and Rural Affairs, 2010a). There are 25 wastewater treatment plants within a 40 km radius of the site, the majority of which are in the northeast to southeasterly wind sector (Department of the Environment and Rural Affairs, 2010b). A landfill site lies 30 km to the east of the site.

TAC is a rural UK site located near the east coast of England. It is 16 km southwest of Norwich (population 200,000), and 28 km east of Thetford (population 20,000), in Norfolk, UK (Office for National Statistics, 2012). Land surrounding the tower is primarily used for agriculture, which is dominated by arable farming (Department of the Environment and Rural Affairs, 2010a). There are three landfill sites between 30 and 50 km from the site, the closest being 30 km to the east (NCC,
2013). There is also a poultry litter power station in Eye, 20 km south of the site (Energy Power Resources Ltd., 2013).

TTA is a rural UK site located near the east coast of Scotland. It is 10 km north of Dundee (population 148,000 (General Register Office for Scotland, 2013)). Land surrounding the tower is predominantly under agricultural use, primarily livestock farming due to its hilly terrain.

HFD is located in rural East Sussex, 20 km from the coast surrounded by woodland, parkland and agricultural green space. The closest large conurbation, Royal Tunbridge Wells (district population



264,000 (Office for National Statistics, 2012)), is located 17km NNE from the tower, while greater London is 40 km NNE.

BSD is a remote moorland plateau site within the North Yorkshire Moors National Park. It is 25km
NNW of Middlesborough (the closest large urban area, population 139,000 (Office for National Statistics, 2012)) and 30km from the coast.





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

1070





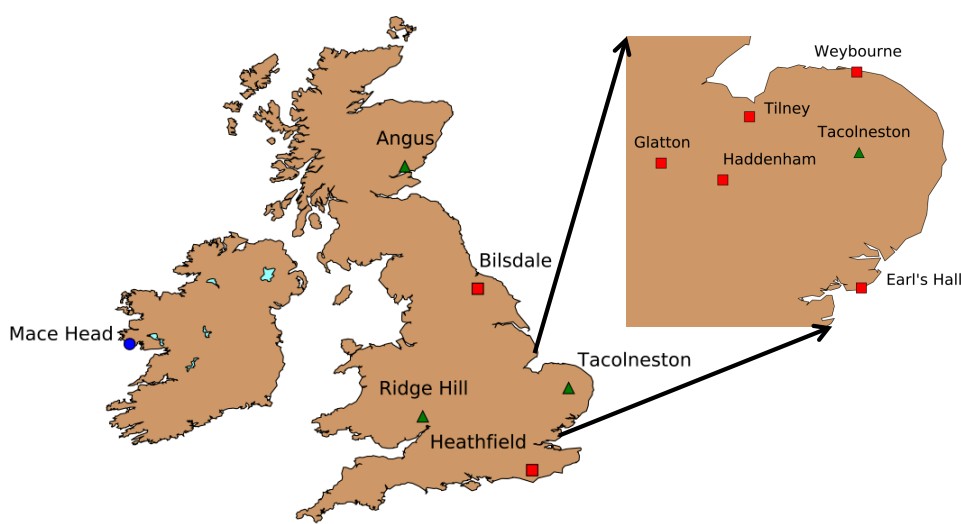

**Figure 1.** The UK DECC network funded by the UK government (sites denoted by green triangles, 2012–), the NERC GAUGE project (denoted by red squares, 2013–2015) and other (blue circle). Sites are described in Table 1 and Appendix A. The enlarged geographical region over East Anglia shows the Church network. These sites are described in Table 4.





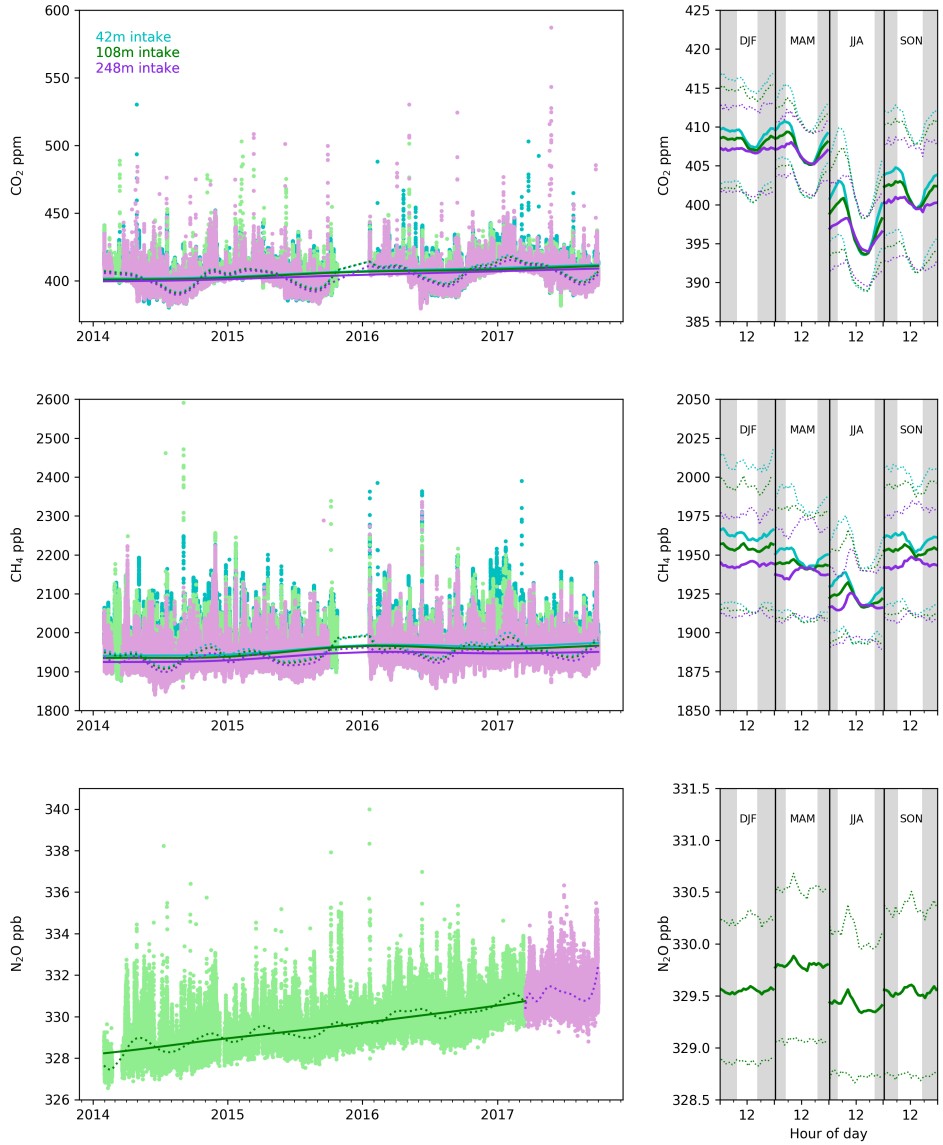

**Figure 2.** Left panels: one-minute mean of $CO_2$ (ppm), $CH_4$ (ppb), and $N_2O$ (ppb) measurements at three inlet heights (42 m, 108 m, and 248 m) at Bilsdale, North Yorkshire from March 2014 to July 2017 (Table 1). The statistical baseline (dashed line) and the long-term trend (solid line) are shown inset for each inlet height. Right panels: mean seasonal diurnal cycle for $CO_2$, $CH_4$, and CO. The dotted lines denote the ±5th and 95th percentile. Statistical fitting procedures follow Thoning et al. (1989); further details can be found in ARS18a.





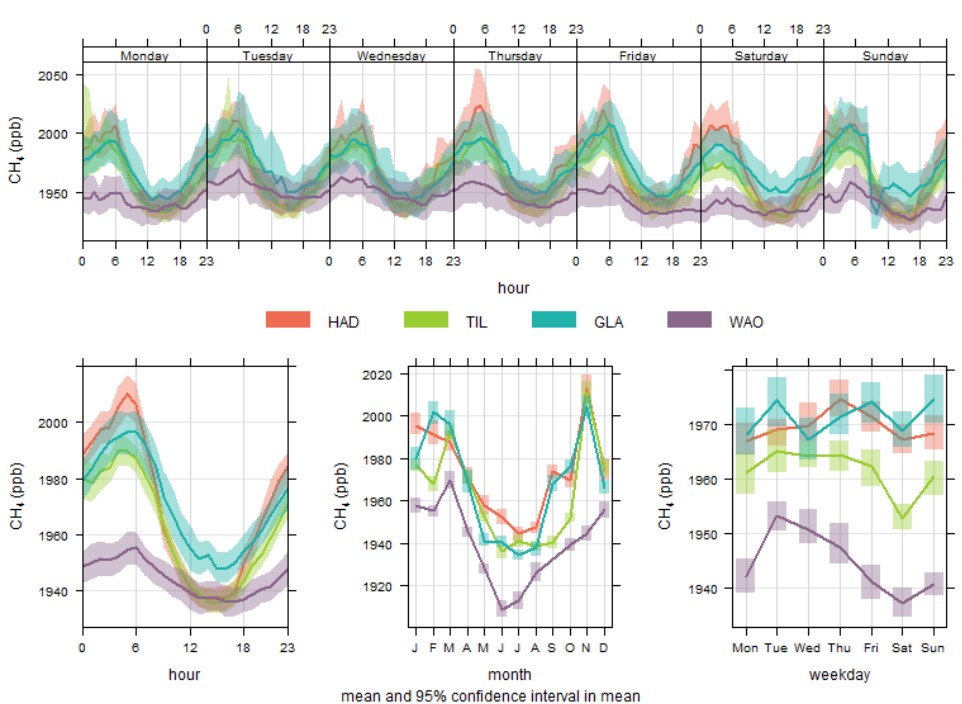

**Figure 3.** Observed variations of $CH_4$ mole fraction data collected at one atmospheric observatory (Weyborne, WAO, 13/2/13–6/5/14), and three church steeples at Haddenham (HAD, 3/7/12–23/9/15), Tilney (TIL, 7/6/13–31/8/15), and Glatton (GLA, 22/10/14–5/4/16). The coloured envelope denotes the 95% confidence interval of the hourly, daily, and monthly mean.


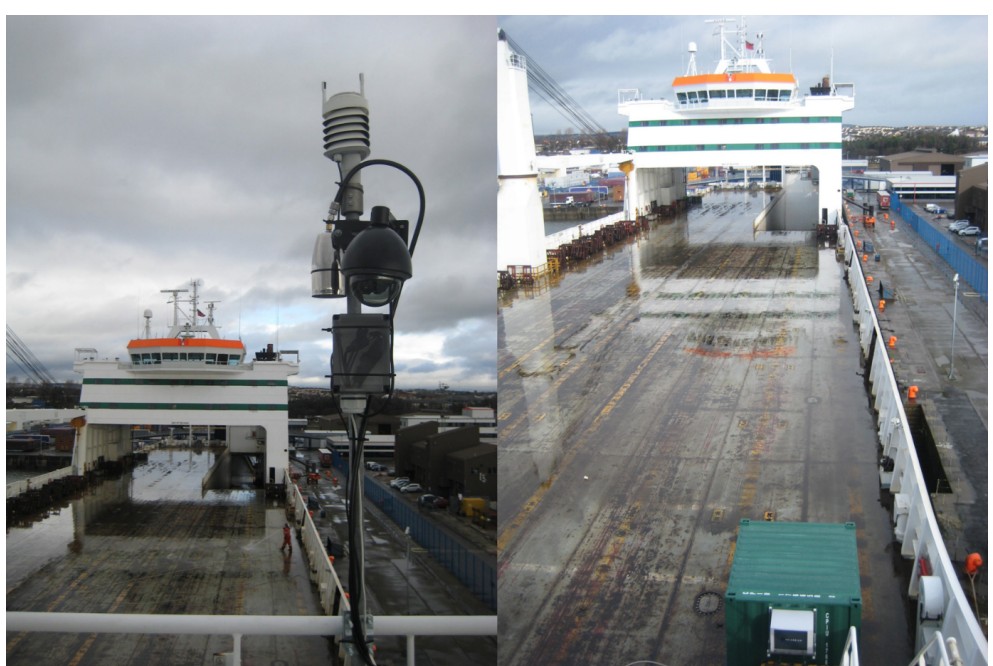

**Figure 4.** Photos of the North Sea ferry mobile GHG laboratory on the DFDS Seaways Longstone (now the Finnmerchant). View of the (left) weather station mounted on the top deck and (right) from the air inlet mounted on top of the mobile laboratory located on the weather deck.




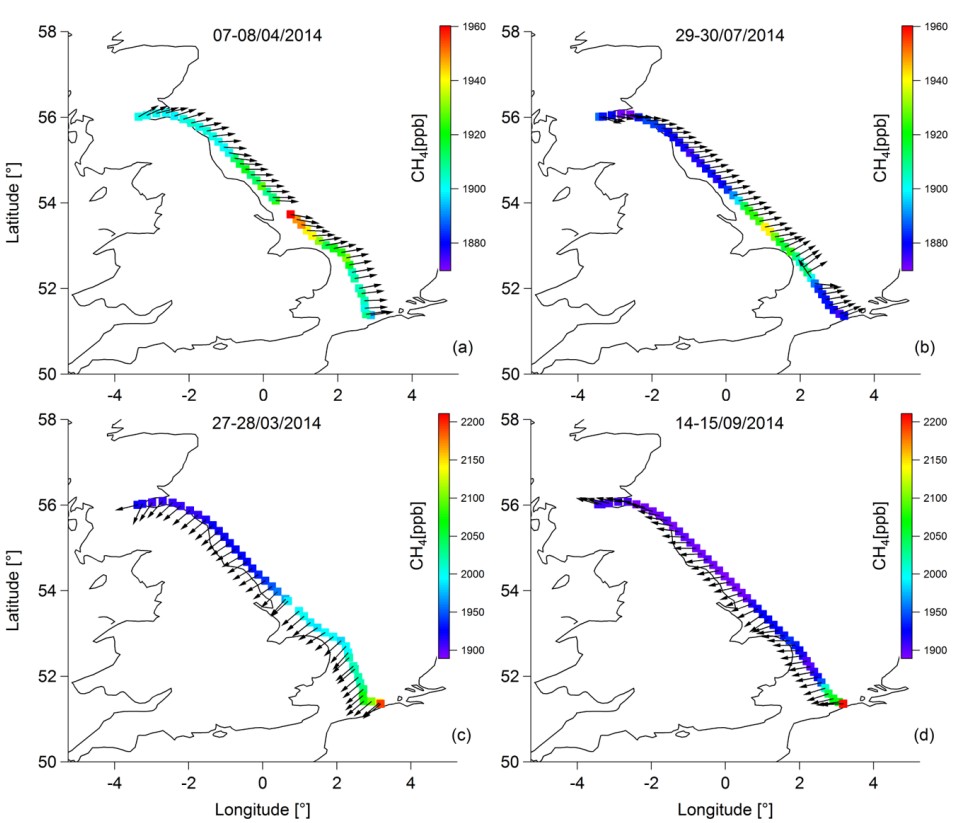

**Figure 5.** Observed temporal and spatial variations in CH$_4$ mole fractions along the route of the DFDS freight ferry in March, April, July and August 2014. Arrows denote local wind direction.





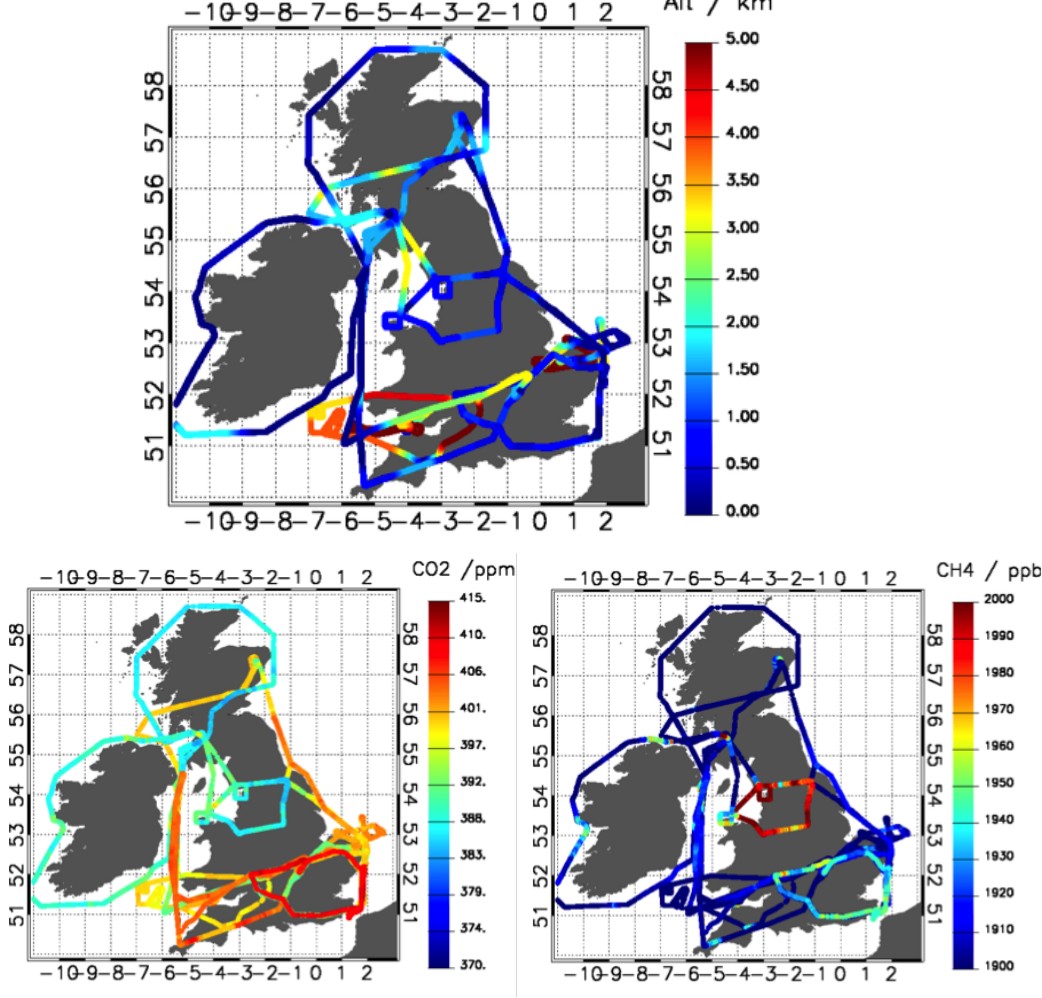

**Figure 6.** Flight tracks for all FAAM flights during GAUGE from 15th May 2014 to 4th April 2016 (Table 6).
Colours denote (top) altitude, (bottom left) $CO_2$ mole fraction, and (bottom right) $CH_4$ mole fraction. See Table
7 for coincidence comparison between the aircraft and tall tower data.





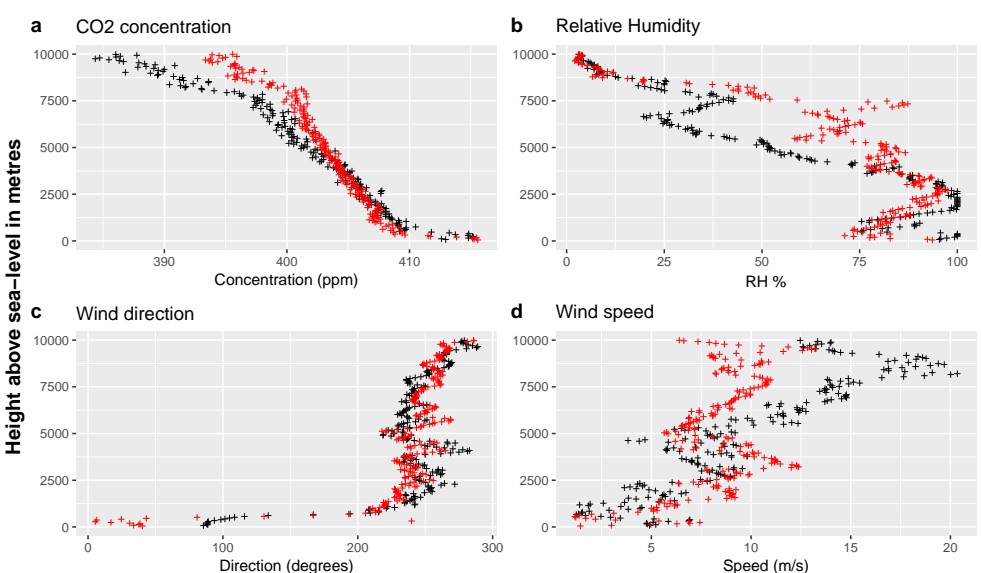

**Figure 7.** Preliminary balloon-borne $CO_2$ data launched on 14th April, 2016 from Weybourne Atmospheric Observatory UK (Figure 1). Correlative measurements of b) relative humidity, c) windspeed and d) wind direction are also shown. Data are averaged every 10 seconds. Red ticks denote the morning launch and black ticks denote the afternoon launch.





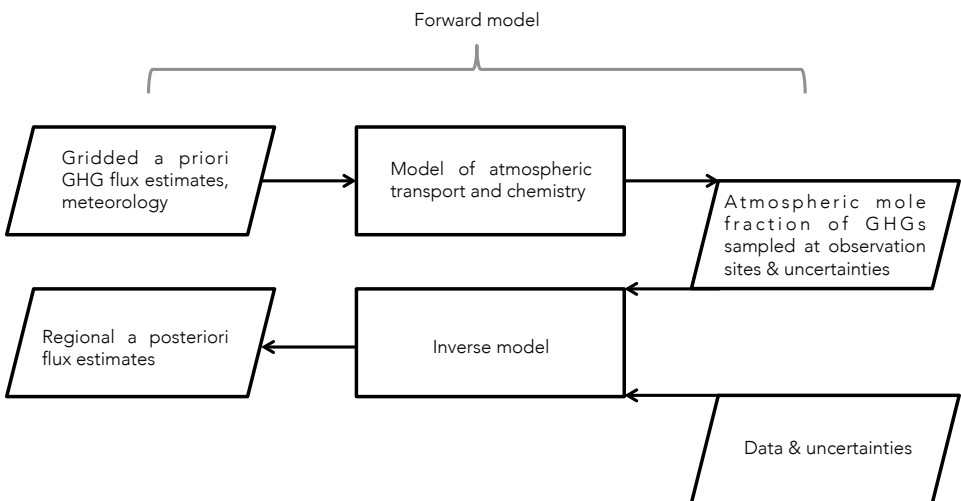

**Figure 8.** Schematic of the generalized GAUGE modelling strategy. The diagram neglects the non-linear inverse modelling approaches.





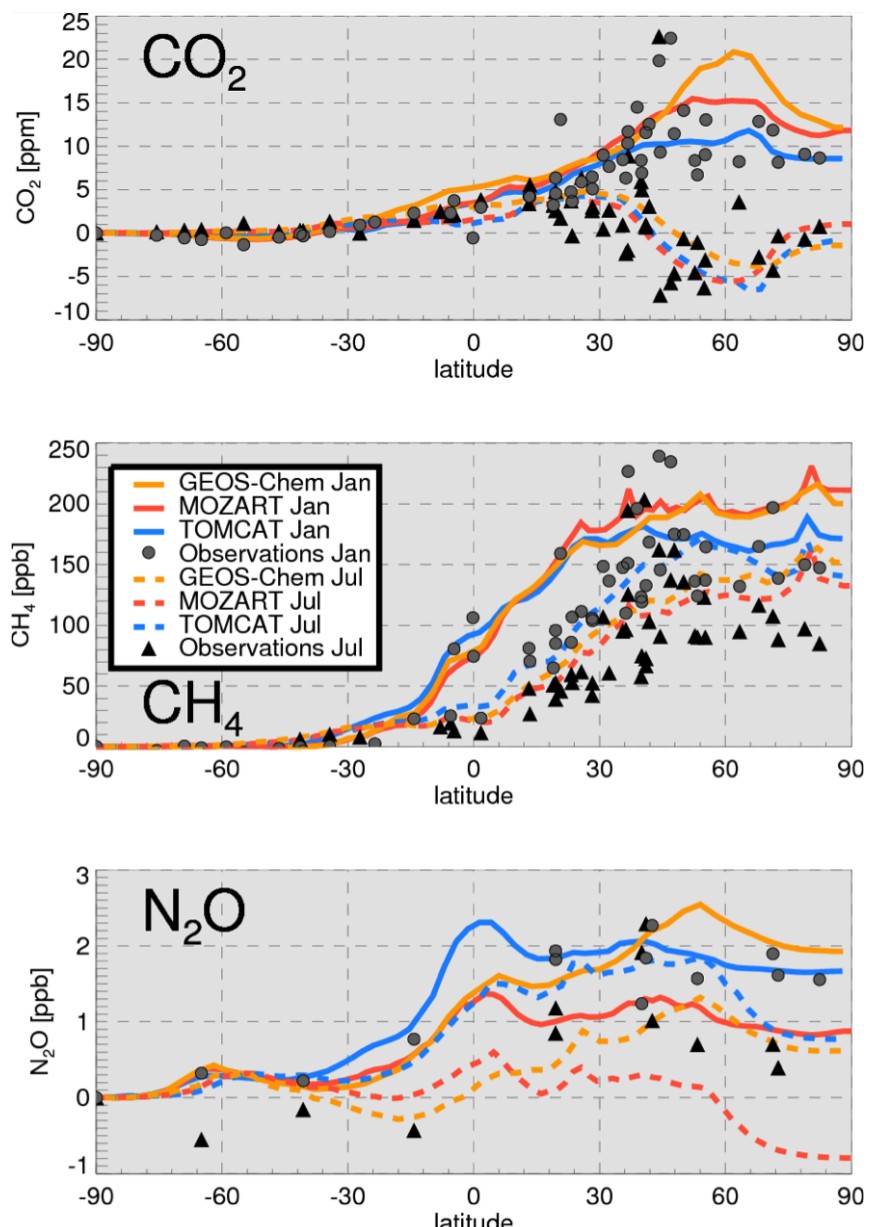

**Figure 9.** Simulated and observed surface zonal mean latitudinal gradient of (a) $CO_2$ (ppm); (b) $CH_4$ (ppb) and (c) $N_2O$ (ppb) in January (solid lines and circles) and July (dashed lines and triangles), 2011. Observations are made as part of NOAA/ESRL measurement campaign. For each model, its South Pole value is subtracted for all latitudes. Observations are treated similarly.





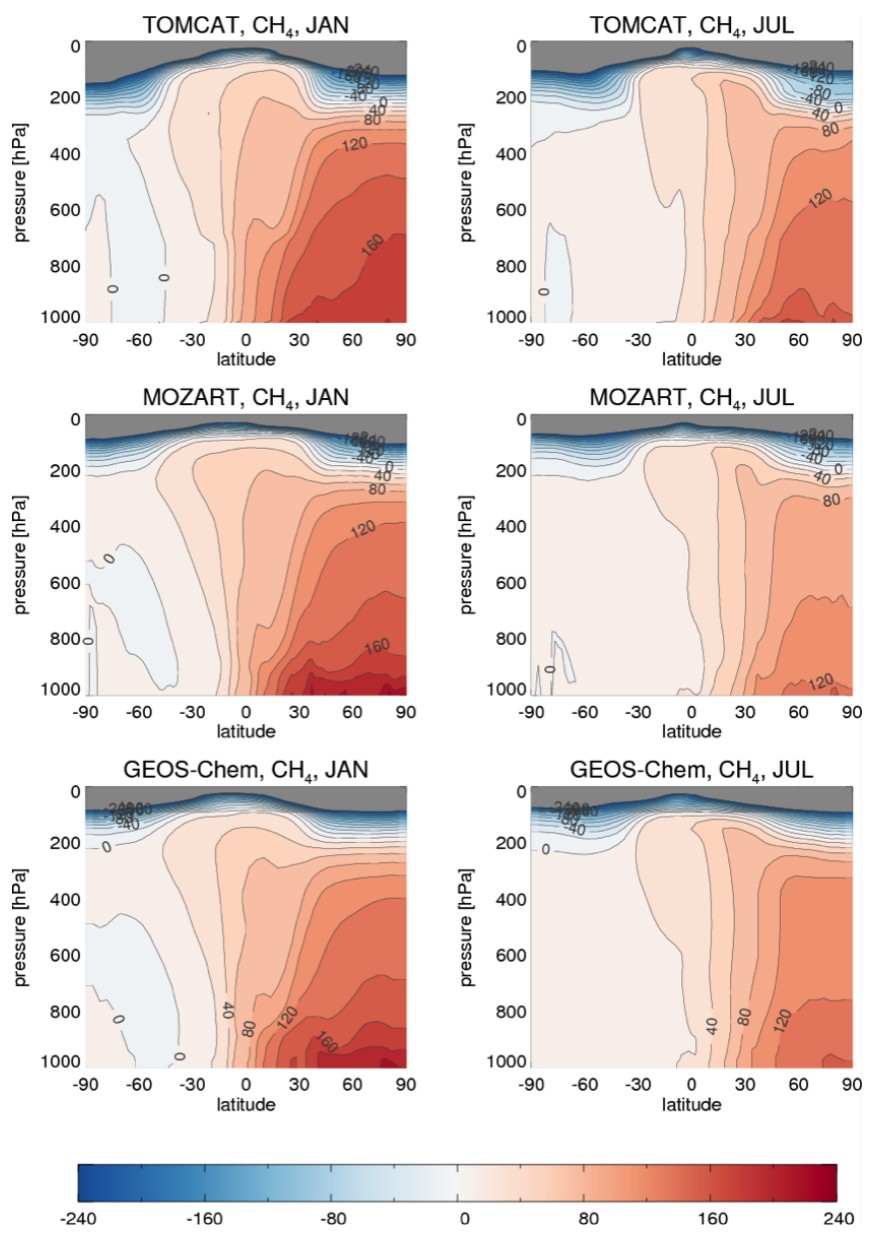

**Figure 10.** Zonal mean distribution of $CH_4$ (ppb) for January (left column) and July (right column) 2011 in each of the GAUGE CTMs. For each model the concentration of $CH_4$ at the surface South Pole concentration is subtracted from the global distribution.

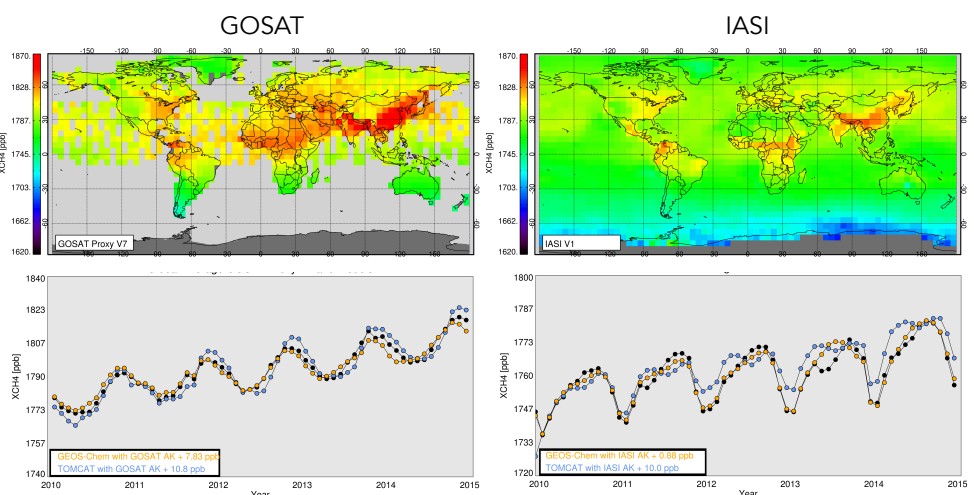

**Figure 11.** Seasonal mean dry air column-averaged mole fractions of $CH_4$ ($XCH_4$) from (top left) GOSAT and (top right) IASI for June-August, 2014, described on a regular $5° \times 5°$ grid. The bottom rows a global mean time series of $XCH_4$ 2010–2015. The GEOS-Chem and TOMCAT models have been sampled at the time and location of individual measurements and convolved with scene-dependent averaging kernels prior to calculating the mean value.





**Table 1.** The name, location, and inlet heights of the UK tall tower network. Entries denoted by an asterisk denote an intake used by a GC-MD and, if present at site, by a Medusa GC-MS.

| Site Name | Acronym | Location | Start/End Date | Altitude (m.a.s.l.) | Inlet Heights (m.a.g.l.) |
|---|---|---|---|---|---|
| Mace Head | MHD | 53.327°N 9.904°W | 23/01/87– | 4 | 10* |
| Ridge Hill | RGL | 51.998°N 2.540°W | 23/02/11– | 204 | 45 & 90* |
| Tacolneston | TAC | 52.518°N 1.139°E | 26/07/11– | 56 | 54, 100* & 185 |
| Angus | TTA | 56.555°N 2.986°W | 13/05/11–29/09/15 | 400 | 222 |
| Bilsdale | BSD | 54.359°N 1.150°W | 30/01/14– | 380 | 42, 108* & 248 |
| Heathfield | HFD | 50.977°N 0.231°E | 20/11/13– | 150 | 50 & 100* |



**Table 2.** Greenhouse gas and ozone depleting substance species and instrumentation at each UK DECC site.

| Species | MHD | TAC | RGL | TTA | BIL | HFD |
|---|---|---|---|---|---|---|
| $CO_2$ | Picarro 2301 | Picarro 2301 | Picarro 2301 | Picarro 2301 | Picarro 2401 | Picarro 2401 |
| $CH_4$ | GC-FID | Picarro 2301 | Picarro 2301 | Picarro 2301 | Picarro 2401 | Picarro 2401 |
| CO | GC-RGA3 | GC-PP1 | — | — | Picarro 2401 | Picarro 2401 |
| $N_2O$ | GC-ECD | GC-ECD | GC-ECD | — | GC-ECD | GC-ECD |
| $SF_6$ | Medusa GC-MS | GC-ECD | GC-ECD | — | GC-ECD | GC-ECD |
| | | Medusa GC-MS | | | | |
| $H_2$ | GC-RGA3 | GC-PP1 | — | — | — | — |
| CRDS Nafion drying period | Cryodried, no nafion | Start−19/6/15 | Start−6/6/15 | 11/1/14−End | Start−1/10/15 | Start−17/6/15 |




**Table 3.** Mean seasonal amplitude and mean growth rates of $CO_2$, $CH_4$ and $N_2O$ at the Bilsdale (BSD), Heathfield (HFD), Ridge Hill (RGL), Tacolneston (TAC), and Angus (TTA) tall tower sites. The mean seasonal amplitude ($\pm 1$ standard deviation) was calculated from the annual peak-to-peak amplitudes. The mean growth rate is the average of the first derivative of the statistical long-term trend.

|  | Site | Intake Height (m) | Mean seasonal amplitude (ppm) | Mean growth rate (ppm/yr) |
|---|---|---|---|---|
| $CO_2$ | BSD | 42 | 18±2 | 3 |
|  |  | 108 | 18±1 | 3 |
|  |  | 248 | 18±1 | 3 |
|  | HFD | 50 | 11±6 | 3 |
|  |  | 100 | 13±5 | 3 |
|  | RGL | 45 | 16±2 | 3 |
|  |  | 90 | 17±2 | 3 |
|  | TAC | 54 | 17±2 | 3 |
|  |  | 100 | 18±2 | 3 |
|  |  | 185 | 18±2 | 2 |
|  | TTA | 222 | 16±1 | 2 |
| $CH_4$ | BSD | 42 | 57±7 | 8 |
|  |  | 108 | 56±2 | 8 |
|  |  | 248 | 41±4 | 7 |
|  | HFD | 50 | 70±40 | 6 |
|  |  | 100 | 60±10 | 7 |
|  | RGL | 45 | 70±20 | 8 |
|  |  | 90 | 60±10 | 8 |
|  | TAC | 54 | 70±20 | 9 |
|  |  | 100 | 70±20 | 9 |
|  |  | 185 | 60±10 | 8 |
|  | TTA | 222 | 31±9 | 13 |
| $N_2O$ | BSD | 108 | 0.8±0.3 | 0.8 |
|  | HFD | 100 | 1.0±0.4 | 0.9 |
|  | RGL | 90 | 1.2±0.3 | 0.9 |
|  | TAC | 100 | 0.6±0.3 | 1.0 |



**Table 4.** Details of the measurements made in the GAUGE East Anglian network.

| Site | Lat [°N], Lon [°E] | Site elevation [m] | Inlet height [m] | Start | End | Measurements | Compounds | Institute lead |
|------|--------------------|--------------------|-------------------|-------|-----|--------------|-----------|----------------|
| Haddenham (HAD) | 52.359, 0.148 | 40 | 25 | 07/2012 | ongoing | GC-FID | $CH_4$ | UCAM |
| Weybourne (WEY) | 52.950, 1.122 | 15 | 15 | 02/2013 | ongoing | GC-FID | $CH_4$, $N_2O$ | UCAM/UEA |
| Tilney (TIL) | 52.737, 0.321 | 6 | 25 | 06/2013 | ongoing | GC-FID | $CH_4$ | UCAM |
| Glatton (GLA) | 52.461, -0.304 | 28 | 20 | 10/2014 | 04/2016 | $in\ situ$ FTIR | $CH_4$, $CO_2$, $N_2O$, CO | ULeic |
| Earls Hall (ELH) | 51.813, 1.118 | 17 | 50 | 11/2014 | 12/2015 | CRDS/QCL | $CH_4$, $CO_2$, $N_2O$ | UCAM |



**Table 5.** Key instrumentation on board the FAAM aircraft for GAUGE-specific flights, including measurement principles and references to instrument characteristics (where available).

| Parameter | Technique | Manufacturer/Model | Reference |
|---|---|---|---|
| CO | VUV Fluorescence | Aerolaser, AL5002 | Gerbig et al. (1999) |
| $O_3$ | UV absorption | Thermo Electron Corporation, 49C | |
| $CH_4$, $CO_2$ | Off axis-integrated cavity Output spectroscopy | Los Gatos, FGGA 907-0010 | O'Shea et al. (2013) |
| $N_2O$, $CH_4$ | Tunable Infrared Laser Differential Absorption Spectroscopy | Aerodyne Research, QC-TILDAS-CS | Pitt et al. (2016) |
| $NO_x$ | Chemiluminescence | Air Quality Design | Di Carlo et al. (2013) |
| HFCs, PFCs, $SF_6$, $C_2$–$C_7$ VOCs | Whole air sampling | Thames Restek | Lewis et al. (2013) |
| $\Delta^{14}CO_2$ | Glass flask sampling | NORMAG | |
| $\delta^{13}CH_4$ | Tedlar bag sampling | SKC | |
| $CO_2$, $CH_4$, $O_3$, $H_2O$, CO | FTIR total column remote sensing | UK Met Office, ARIES | Allen et al. (2014) |
| Humidity | Chilled Mirror | General Eastern, GE 1011B | Ström et al. (1994) |
| Temperature | PRT | Rosemount Aerospace, 102 AL | Petersen and Renfrew (2009) |
| Wind vector | 5-hole probe | BAE Systems & UK Met Office | Brown et al. (1983) |





**Table 6.** Diary of FAAM survey flights for GAUGE between May 2015 and March 2016, including take-off and landing times; and sampling locations and brief description of mission profiles.

| Flight No. | Date | Take-off (UTC) | Landing (UTC) | Description |
|---|---|---|---|---|
| B848 | 15/05/14 | 12:07:07 | 16:46:25 | North Sea Gas Rigs (+instrument test flight) |
| B849 | 16/05/14 | 09:33:16 | 12:45:28 | Bristol Channel (+instrument test flight) |
| B850 | 21/05/14 | 07:59:54 | 15:22:59 | Around Britain – UK outflow |
| B851 | 17/06/14 | 09:56:43 | 14:43:25 | Southwest Approaches – UK inflow |
| B852 | 18/06/14 | 08:25:01 | 16:29:35 | Around Britain – DECC Tower survey |
| B861 | 09/07/14 | 08:55:32 | 13:20:52 | Around London – mass balancing |
| B862 | 15/07/14 | 10:59:32 | 15:17:35 | Around London – mass balancing |
| B864 | 01/09/14 | 08:09:57 | 10:49:27 | Irish Sea – transit to Prestwick |
| B865 | 01/09/14 | 13:03:45 | 15:51:41 | Around Scotland - – mass balancing |
| B866 | 02/09/14 | 08:08:16 | 12:01:38 | Around Ireland – mass balancing |
| B867 | 02/09/14 | 13:24:29 | 17:11:09 | Around Ireland – area survey |
| B868 | 04/09/14 | 11:57:58 | 16:40:22 | Northwest England – sources of 14C |
| B905 | 12/05/15 | 07:59:00 | 11:34:02 | Irish Sea SW Approaches – upwind of UK |
| B906 | 12/05/15 | 13:09:14 | 17:03:19 | North Sea – UK outflow |
| B911 | 28/05/15 | 07:55:04 | 10:19:26 | Around Britain – aborted (instrument fault) |
| B948 | 04/03/16 | 08:55:20 | 14:10:19 | Around London – mass balancing |



**Table 7.** Comparison between aircraft and ground-based one-minute mean measurements for $CO_2$ and $CH_4$ during tall tower fly pasts.

| Flight | Station | Inlet altitude (m a.s.l.) | Aircraft Altitude (m a.s.l.) | Tall tower data $CO_2$ (ppm) | $CH_4$ (ppb) | Aircraft minus tall tower data $CO_2$ (ppm) | $CH_4$ (ppb) |
|---|---|---|---|---|---|---|---|
| B850 | TTA | 522 | 1022 | 399.38 | 1913.99 | 1.39 | 5.50 |
| B852 | TTA | 522 | 1039 | 397.37 | 1908.83 | 2.26 | -58.32 |
|  | BSD | 488 | 738 | 398.05 | 1878.45 | 0.11 | -1.10 |
| B861 | RGL | 290 | 576 | 390.22 | 1912.52 | 2.50 | -1.48 |
|  | HFD | 250 | 687 | 402.17 | 1945.77 | -0.23 | -9.49 |
|  | TAC | 104 | 683 | 388.57 | 1876.77 | 5.86 | 12.28 |
|  | RGL | 290 | 488 | 390.88 | 1901.85 | -0.15 | 1.34 |
|  | HFD | 250 | 465 | 395.18 | 1920.95 | 1.07 | -1.38 |
|  | TAC | 104 | 675 | 390.66 | 1876.53 | 4.53 | 13.71 |
| B862 | RGL | 290 | 440 | 388.84 | 1891.37 | 1.00 | -4.30 |
|  | HFD | 200 | 425 | 392.88 | 1869.02 | -0.14 | 1.22 |
|  | RGL | 290 | 482 | 389.31 | 1886.66 | -0.25 | 1.71 |
|  | HFD | 200 | 690 | 392.93 | 1874.60 | 0.77 | 0.70 |
|  | TAC | 235 | 373 | 393.56 | 1900.04 | -3.32 | -6.64 |
| B865 | TTA | 522 | 687 | 384.72 | 1891.21 | 0.09 | 1.37 |
| B866 | MHD | 29 | 119 | 387.63 | 1891.53 | 0.34 | 3.54 |
| B867 | MHD | 29 | 105 | 389.34 | 1913.09 | 1.48 | 11.91 |
| B868 | BSD | 628 | 708 | 386.32 | 1987.75 | 0.13 | -0.51 |
|  | BSD | 628 | 569 | 384.82 | 1980.36 | 0.25 | -0.81 |
| B905 | RGL | 290 | 431 | 404.32 | 1911.47 | -0.40 | 3.48 |
|  | RGL | 245 | 457 | 401.83 | 1909.53 | 1.48 | -1.06 |
| B906 | TAC | 104 | 275 | 401.13 | 1912.96 | 0.13 | -2.36 |
|  | TAC | 150 | 296 | 399.85 | 1913.34 | 0.09 | 1.04 |
| B911 | RGL | 290 | 713 | 403.49 | 1914.23 | 0.62 | 1.62 |
| B948 | RGL | 245 | 405 | 409.10 | 1946.96 | 0.61 | -2.00 |
|  | HFD | 250 | 346 | 411.39 | 1951.87 | -0.86 | -2.84 |
|  | TAC | 104 | 406 | 408.86 | 1942.21 | 0.07 | -0.04 |
| **Average** |  |  |  |  |  | **0.72** | **-1.22** |
| **Standard Deviation** |  |  |  |  |  | **1.69** | **12.54** |





**Table 8.** Summary of successful ChemSonde launches from Weybourne Atmospheric Observatory.

| Instrument # | Launch date/time (UTC) | Altitude/Distance | Comments |
| --- | --- | --- | --- |
| 709D | 14th April, 10:39 | 33.5/40.2 km | Successful launch: $CO_2$ and meteorology data. |
| 7097 | 14th April, 14:30 | 34.2/43.3 km | Successful launch: $CO_2$ and meteorological data. |

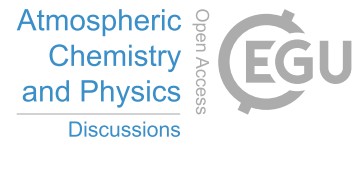

**Table 9.** Model descriptions used in the GAUGE inter-comparison.

| Model | Institute | Forward Model Type | Horizontal (Nested) Resolution | Vertical Resolution | Meteorology | Inverse Method | Key References |
|---|---|---|---|---|---|---|---|
| GEOS-Chem | U. Edinburgh | E | 2°×2.5° (0.25°×0.3125°) | 47 levels (surface to 0.01 hPa) | NASA GEOS-5 | EnKF | Feng et al. (2009, 2011, 2017) |
| MOZART | U. Bristol | E | 2.5°×1.9° | 56 levels (surface to 1.65 hPa) | NASA GEOS-5 | 4D-Var | Emmons et al. (2010) |
| TOMCAT | U. Leeds | E | 1.125°×1.125° | 60 levels (surface to 0.1 hPa) | ECMWF ERA Interim | 4D-Var | Wilson et al. (2014); McNorton et al. (2016); Monks et al. |
| NAME | Met Office/ U. Bristol | L | 1.5 km over UK domain | 60 levels (surface to 29 km) | Met. Office | Bayesian inference | Manning et al. (2011); Ganesan et al. (2015) |