# Peer review of "A measurement-based verification framework for UK greenhouse gas emissions: an overview of the Greenhouse gAs Uk and Global Emissions (GAUGE) project"

_Atmospheric Chemistry and Physics, 2018_

## Referee Comment (RC1) · Anonymous Referee #1 · 2 Apr 2018

Overall comments:

This overview paper of the GAUGE project generally serves its purpose. We find that a lot of focus is put on describing the measurements and the global models, and as can be found below, I think some connection between the two might be nice to tie the two together, otherwise the paper is a bit disjointed. There is very little reference to how the models will use the measurements (and which measurements), which I think is needed to put some context onto both.

[Figure]

Also, there are a lot of references to material in prep, not ideal but understandable given the overview nature of this manuscript.

Specifically:

L 89 No comma after "Although"

L183 - Should be mole fractions of Co2 have a peak diurnal cycle of 10 ppm - these are not flux units.

L202 - please double check the flask information here - I am not sure, but I thought that MHD is part of the surface flask network which does not use PFP's (Hermes) but rather the Sherpa. But perhaps this is not the case for the C14 samples?

http://www.hpd-online.com/air-samplers.php

L205-210 - what about TAC, were those samples collected & measured the same way as MHD? If not, uncertainties in the differences might be larger?

L246-250 and Figure 3: This is a very nice looking figure, but it seems that one must account for inter-annual variability (in the global methane concentrations as well as local fluxes) when comparing two sites with different sampling time frames. Sites that sampled only in 2013-2014 (WAO) are compared directly presumably with averages from other sites over different time periods (GLA, 2014-2016?). Some sort of normalization must occur here, or restricting the comparisons to averages over a common time period - also it seems that some sites might have sampled more in different seasons than others (i.e. can you really compare an average diurnal cycle if one site sampled 3 summers and two winters vs. 2 summers and 3 winters)?

L290 - Figure 5 does not have (a), (b) indicated, but they are referred to here. I would recommend a little more discussion of how ship emissions were avoided or perhaps removed later in filtering. It seems unlikely that no ship exhaust was ever measured given the inlet is not at the bow?

L341/Table 7 discussion: Do these numbers tell you anything about uncertainty in each measurement? As the authors state, clearly some vertical differences are to be expected but it seems that perhaps the time periods could be filtered based on conditions (higher winds, higher PBLs, etc) that one would expect there to be little vertical gradient so that an uncertainty could be determined from this comparison. Otherwise it seems that there was not much point to the comparison. (or at least perhaps it would give us an upper limit on the uncertainty or bias?).

L345 Wording awkward at the end - what is more sensitive to local fluxes?

L350. Perhaps details of the SenseAir NDIR sensor will be in another publication (although I don't see a citation here), but at least a model number would allow us to look up the specific sensor used here, as SenseAir makes several? Or is this something custom made for the ChemSonde? Certainly this method looks very promising. Is data only collected on the ascent? If both, then is there relatively good agreement between ascending and descending data from the same sensor? Are the sensors/payloads recovered or considered expendable?

L407: What kind of co2 sensor is on the UAV? (commercial or custom sensor)? Later CH4 is mentioned also from UAV.

L503: There are other good reasons to run more than one model than just to mitigate criticism! (Gives an idea of transport model uncertainty and spread of possible results, for one (as noted in L519)). This should probably be rephrased.

L515- Regarding the NAME model, this unlike the others is a dispersion model - this could be pointed out and identify what the underlying transport model is used with it.

L635 - the NAME model and inverse method is not given as much space as the global models here, so some questions remain - what is the domain of this inversion, presumably regional rather than global? Was the forward model evaluated as it was for the global models?

For all the models it would be nice to see comparisons (of the forward runs perhaps) with the observations that are in the beginning of the text, assuming these observations are used in the inversions. Three figures are dedicated to evaluating the basic transport of the global models with comparisons to observations that are not part of GAUGE. It would be nice to link the two sections of this manuscript: the observations and the models in some fashion here, because it reads a bit disjointed.

―――――――――――――――――――――

---

## Referee Comment (RC2) · Anonymous Referee #2 · 5 Apr 2018

The manuscript "A measurement-based verification framework for UK greenhouse gas emissions: an overview of the Greenhouse gAs Uk and Global Emissions (GAUGE) project" by P. Palmer et al. presents the motivation, design and execution of a research project aimed at quantifying the UK budget of the major greenhouse gases ($CO_2$, $CH_4$ and $N_2O$) over the period 2013-2015. The paper describes on the one hand the measurement strategy, consisting of various types of observations, adapted for the project in order to achieve its goal of a sectorial GhG quantification. On the other hand it outlines the project's modelling strategy that should make use of the measurements to

estimate the magnitude, distribution and uncertainty of the UK GhG emissions.

Overall the manuscript is concise and well written. provides a nice overview of the GAUGE project and serves very well as an introductory paper to the special issue. However, this is exactly the main criticism: the paper does not mention at all that it belongs to the special issue nor that it actually stands as an introductory article for this special issue. Taken as a stand alone paper it is in fact rather weak on the science; it mentions different measurement strategies and different modelling approaches but it hardly provides any analysis of results but refers to dedicated papers for these results. This would be fine if the paper would be set in context to the special issue.

The paper would also benefit from a more in depth discussion in the concluding remarks especially with regard to providing advice to the community for planning and setting up future GhG monitoring and quantification systems, for instance, which measurements were helpful, which did not provide extra information, which modelling strategy seemed to be more successful and why etc.

Detailed comments: L 110: Lower posterior fluxes than prior fluxes doesn't mean that they are necessarily better or more correct. Is there a way to qualify this?

Section 2.2 North Sea Ferry: I am not sure about the nomenclature here, but I would not consider a ferry with a fixed route as a mobile platform. A mobile platform is a platform that can be moved to different places depending on external circumstances, which is not the case with a fixed ferry route.

L 304: 'flux inversion models' is often wrongly used as a term to refer to inverse modelling system solving for fluxes. These systems are inverting atmospheric concentrations and yielding fluxes as a result of the inversion. So better to use just 'inversion models' or 'inverse modelling system'.

L 495: Which sectors are described?

Ll 499/500: Do you also take into account model uncertainty, and how do you quantify

the model uncertainty?

L 571: Where do the boundary conditions come from?

Ll 531ff: I think you need to comment in the paper on the error you make when using anthropogenic CO2 emissions from 2009 for the years beyond 2009. The same is true for the CH4 emissions beyond 2010. How can climatological ocean fluxes cover a certain period?

Ll 562ff: You need to comment on the fairly large model spread and how this effects the quantification of the emissions in the inversion. It would also be helpful to provide the spread in relative units to better understand the differences in the spread between the three gases.

L 577: Do you mean Jan instead of July here?

Ll 589ff: You don't mention MOZART here, is there a specific reason for this?

Ll 598/599: I don't understand how these biases reflect errors in the prior emissions if you use the same prior for both models.

Ll 605/605: How do the different methods impact on the resulting posterior fluxes (in addition to the different forward models)?

Ll 618/619: Can you quantify this or provide a reference for this statement.

Ll 651ff: It would be nice if you could give an example of estimated fluxes here as well and not only refer to other papers in the special issue.

Ll 698ff: Is the coverage also too sparse for estimating CH4 fluxes? I would imagine that it is not only a matter of higher spatial resolution but also depends on the revisit time to 'see' more cloud free scenes.

Fig 2: It seems to be not very useful to display one-minute mean values over 4 years, the individual values are not visible at all. Maybe aggregate the measurements to lower

[Figure]

temporal resolution or show a much shorter time period.

Fig 9 top panel: Where does the 'outlier' with a value of ∼23 ppm in the Jul observations at approximately 40 degrees come from?

Fig 11: The different lines in the time series plot for GOSAT are hardly visible.

Tab 8: This table doesn't convey much information and could be removed.

Tab 9: 'E' and 'L' stand for Eulerian and Lagrangian model type, please explain in the table caption.

---

## Referee Comment (RC3) · Anonymous Referee #3 · 10 Apr 2018

The paper gives an overview of the GAUGE project, which combines atmospheric observations of greenhouse gases to quantify the UK GHG budget. In general the paper is well written, and I recommend publication after the following, mostly minor concerns have been addressed.

General Comments:

The introduction should refer to other networks, such as the global greenhouse gas reference network from NOAA ESRL GMD, the research infrastructure ICOS, or the

Integrated Global Greenhouse Gas Information System IG3IS.

In this overview paper many papers are referenced as "in preparation for ACP" (Stavert et al., Wenger et al., Connors et al., Helfter et al., Pitt et al., Lunt et al., Palmer et al.), is it the intention to have these published soon so that they can be properly referred to? This would be very helpful.

Balloon CO2 sondes: The section seems a bit speculative. The two profiles (morning and afternoon) start deviating strongly (1-10 ppm) above about 4 km, while the description in the text mention problems with baseline drift and span measurement during one of the flights affecting only data collected above 6.5 km. Given that the traceability requirements for using such data for validation of space-borne remote sensing of GHG measurements is quite tight, the balloon CO2 sonde is far away from being useful. I suggest shortening the section, just pointing out the status of the system.

Intercalibration activities: As multiple calibration scales are used for CH4 and for N2O, it should be made clear that for use in (inverse) modelling the data need to be put on the same scale to reduce any impact from bias errors.

Specific comments

Table 1: Please explain "CG-MD"

L 177: The Thoning et al. (1989) method does not provide a baseline, it is rather a curve fitting using harmonics and trend. A baseline would need to have some filtering of e.g. polluted episodes to retain the "baseline conditions".

Fig. 2, left panels: the colours used for the time series plots differ quite a bit from those in the legend. I suggest adopting the legend colours for the plotting.

L 186: the role of the boundary layer height during winter should be explained a bit more.

L 202: "We added an extra flask to the collection" this is unclear

[Figure]

L 237: suggestion to replace "to" with "and"

L 250: "boundary layer likely plays the dominant role" do you mean boundary layer height? Otherwise this is a trivial statement as the boundary layer is sampled by the sites.

L 286: "Differences between sailing ..." does that refer to different trips or directions of the ferry? Should this be "sailings"?

L 345: "and more sensitive" -> "and higher sensitivity"

L 535: replace ";" with "and"

L 560: "free-running CTMs" from Table 9 it seems that all transport models use assimilated meteorological fields

---

## Author Comment (AC1) · 15 Jun 2018

We thank the three reviewers for providing useful comments to improve our paper. Below we respond to each reviewer comment individually with the original comment denoted by italics.

**Reviewer 1**

*There is very little reference to how the models will use the measurements (and which measurements), which I think is needed to put some context onto both.*

[Figure]

We have added some additional text to augment the submitted manuscript. In general, the models used all the data.

*L 89 No comma after "Although"*

Done.

*L183 - Should be mole fractions of Co2 have a peak diurnal cycle of 10 ppm - these are not flux units.*

We thank the review for spotting this typo. These numbers refer to mole fractions not fluxes.

*L202 - please double check the flask information here - I am not sure, but I thought that MHD is part of the surface flask network which does not use PFP's (Hermes) but rather the Sherpa. But perhaps this is not the case for the C14 samples? http://www.hpd-online.com/air-samplers.php*

The reviewer is correct that the sample module is a Sherpa not a Hermes. We have amended the text.

*L205-210 - what about TAC, were those samples collected  measured the same way as MHD? If not, uncertainties in the differences might be larger?*

The sample collection at TAC is based on the same method as a Sherpa (used at MHD) and sent for analysis to NOAA/INSTARR (same as MHD samples), so the sample errors between both sites will be very similar.

*L246-250 and Figure 3: This is a very nice looking figure, but it seems that one must account for inter-annual variability (in the global methane concentrations as well as local fluxes) when comparing two sites with different sampling time frames. Sites that sampled only in 2013-2014 (WAO) are compared directly presumably with averages from other sites over different time periods (GLA, 2014-2016?). Some sort of normalization must occur here, or restricting the comparisons to averages over a common time*

*period - also it seems that some sites might have sampled more in different seasons than others (i.e. can you really compare an average diurnal cycle if one site sampled 3 summers and two winters vs. 2 summers and 3 winters)?*

This is a good point. Our text was mainly focused on the size of sub-annual variations that are not necessarily related to inter-annual changes, which may help to explain the offset between WAO and the church sites. We clarify this point this in text.

*L290 - Figure 5 does not have (a), (b) indicated, but they are referred to here. I would recommend a little more discussion of how ship emissions were avoided or perhaps removed later in filtering. It seems unlikely that no ship exhaust was ever measured given the inlet is not at the bow?*

Figure 5 does include labels (bottom right on each panel), but we have redrawn Figure 5 with larger labels. To avoid contamination from GHG emissions onboard the ship (e.g. engine emissions, venting of below deck cargo area) individual data points were removed when the ship was in port or when the wind blew from the direction of the chimney stacks determined using our onboard weather station.

*L341/Table 7 discussion: Do these numbers tell you anything about uncertainty in each measurement? As the authors state, clearly some vertical differences are to be expected but it seems that perhaps the time periods could be filtered based on conditions (higher winds, higher PBLs, etc) that one would expect there to be little vertical gradient so that an uncertainty could be determined from this comparison. Otherwise it seems that there was not much point to the comparison. (or at least perhaps it would give us an upper limit on the uncertainty or bias?).*

On reflection, prompted by this reviewer comment, we have decided that the information shown in Table 7 does not add much to the paper. We refer the reader to the paper that is focused on the analysis of the aircraft data.

*L345 Wording awkward at the end - what is more sensitive to local fluxes?*

Agreed. This sentence contains several grammatical errors. We have removed it based on the previous comment.

*L350. Perhaps details of the SenseAir NDIR sensor will be in another publication (although I don't see a citation here), but at least a model number would allow us to look up the specific sensor used here, as SenseAir makes several? Or is this something custom made for the ChemSonde? Certainly this method looks very promising. Is data only collected on the ascent? If both, then is there relatively good agreement between ascending and descending data from the same sensor? Are the sensors/payloads recovered or considered expendable?*

We have provided additional details associated with these questions.

*L407: What kind of co2 sensor is on the UAV? (commercial or custom sensor)? Later CH4 is mentioned also from UAV.*

The CO2 sensor mentioned in the text is a commercial NDIR sensor, an Edinburgh Instruments Gascard NG. We used a ground-based Los Gatos Research Ultra-portable Greenhouse Gas Analyser to measure atmospheric CH4, connected to the UAV via a 150 m long Teflon inlet. This has been clarified in the text.

*L503: There are other good reasons to run more than one model than just to mitigate criticism! (Gives an idea of transport model uncertainty and spread of possible results, for one (as noted in L519)). This should probably be rephrased.*

Agreed. It was not our intention to give that impression. We have amended the text.

*L515- Regarding the NAME model, this unlike the others is a dispersion model - this could be pointed out and identify what the underlying transport model is used with it.*

Agreed. NAME is a Lagrangian dispersion model and used on a regional basis. For GAUGE, 3-D meteorology from the Met Office numerical weather prediction model provided the flow fields required by NAME. The boundary conditions for NAME were solved as part of the inverse problem.

*L635 - the NAME model and inverse method is not given as much space as the global models here, so some questions remain - what is the domain of this inversion, presumably regional rather than global? Was the forward model evaluated as it was for the global models?*

We have clarified these points in the text. The domain of the regional inversions is stated as being north west Europe. Prior time-series using the prior emissions and prior boundary conditions, akin to a forward simulation, are used directly in the inverse frameworks.

*For all the models it would be nice to see comparisons (of the forward runs perhaps) with the observations that are in the beginning of the text, assuming these observations are used in the inversions. Three figures are dedicated to evaluating the basic transport of the global models with comparisons to observations that are not part of GAUGE. It would be nice to link the two sections of this manuscript: the observations and the models in some fashion here, because it reads a bit disjointed.*

Good suggestion. As part of our preliminary studies we sampled the model for the tall tower sites. We have now included a Figure that shows the results of that comparison at one of those sites.

**Reviewer 2**

*Taken as a stand alone paper it is in fact rather weak on the science; it mentions different measurement strategies and different modelling approaches but it hardly provides any analysis of results but refers to dedicated papers for these results. This would be fine if the paper would be set in context to the special issue.*

Agreed. This is the role of this paper. It is intended to set the scene and provide the overarching details, thereby freeing the science-led papers in the special issue to devote their pages to science.

*The paper would also benefit from a more in depth discussion in the concluding re-*

*marks especially with regard to providing advice to the community for planning and setting up future GhG monitoring and quantification systems, for instance, which measurements were helpful, which did not provide extra information, which modelling strategy seemed to be more successful and why etc.*

This will be included in a later synthesis paper. Science-led papers are still being prepared/submitted. A suggested measurement strategy is emerging but is outside the scope of this overview paper.

*Detailed comments: L 110: Lower posterior fluxes than prior fluxes doesn't mean that they are necessarily better or more correct. Is there a way to qualify this?*

This is a fair point. Ganesan et al 2015 do not appear to qualify their results as better than the prior. However, because they have fitted the estimates to observed atmospheric mole fraction measurements they can say that their posterior fluxes are more consistent with the available data. We will qualify our statement in the manuscript.

*section 2.2 North Sea Ferry: I am not sure about the nomenclature here, but I would not consider a ferry with a fixed route as a mobile platform. A mobile platform is a platform that can be moved to different places depending on external circumstances, which is not the case with a fixed ferry route.*

Fair point. We are happy to re-describe this platform as a moving mobile platform. We have changed the manuscript accordingly.

*L 304: 'flux inversion models' is often wrongly used as a term to refer to inverse modelling system solving for fluxes. These systems are inverting atmospheric concentrations and yielding fluxes as a result of the inversion. So better to use just 'inversion models' or 'inverse modelling system'.*

Fair point. We have changed the manuscript accordingly.

*L 495: Which sectors are described?*

In practice, it is difficult to separate the influence of individual sectors. We have decided to remove the mention of sectors in this statement.

*Ll 499/500: Do you also take into account model uncertainty, and how do you quantify the model uncertainty?*

Yes. Different inverse methods use different approaches. Based on this comment we have added a statement about how different descriptions of the inverse problem can contribute to the spread of posterior flux estimates.

*L 571: Where do the boundary conditions come from?*

We think the reviewer is referring to line 517. Lateral boundary conditions for NAME (the only regional model used within GAUGE) are generally solved as part of the InTEM but can also use archived values from global models. We have clarified this point in the manuscript.

*Ll 531ff: I think you need to comment in the paper on the error you make when using anthropogenic CO2 emissions from 2009 for the years beyond 2009. The same is true for the CH4 emissions beyond 2010. How can climatological ocean fluxes cover a certain period?*

The purpose of this experiment was primarily to determine the agreement between models that are used to infer fluxes from data. Because our knowledge of the carbon cycle is incomplete there was always going to be an accumulated drift in the comparison. Ocean fluxes are admittedly difficult to describe. The underlying assumption associated with using a climatology is that ocean fluxes change on large spatial and temporal scales cf continental source. The alternative approach is to use a model fitted to time-dependent pCO2 data but these measurements are very sparse and do not necessarily constrain model parameters uniformly from one year to the next. In short, there is no perfect way to describe this CO2 flux. Nevertheless, we added some text to justify our approach.

*Ll 562ff: You need to comment on the fairly large model spread and how this effects the quantification of the emissions in the inversion. It would also be helpful to provide the spread in relative units to better understand the differences in the spread between the three gases.*

We have revised the Figure so that it now include the maximum spread between models during January and July. The latitudinal distribution has been normalised to the South Pole value for each model. This was done to account for the drift (incorrect sources/sinks) associated with the eight-year simulation. It looks like the biggest differences are only in the NH, but it also reflects differences in interhemispheric transport times between the models.

*L 577: Do you mean Jan instead of July here?*

Yes. We thank the reviewer for spotting that error.

*Ll 589ff: You don't mention MOZART here, is there a specific reason for this?*

No fundamental reason. The TOMCAT and GEOS-Chem models were already set-up for this analysis, including the necessary observation operators, building on published studies.

*Ll 598/599: I don't understand how these biases reflect errors in the prior emissions if you use the same prior for both models.*

This sentence was not well written and we have chosen to remove it.

*Ll 605/605: How do the different methods impact on the resulting posterior fluxes (in addition to the different forward models)?*

This a good question. They can impact posterior fluxes via, for example, different descriptions of model error and different assumptions associated with the error covariance matrices used by the inversion methods. We have provided a brief description of the different ways this can happen.

*Ll 618/619: Can you quantify this or provide a reference for this statement.*

We have included a reference for CH4 and CO2. https://www.atmos-chem-phys.net/13/9917/2013/ https://www.atmos-chem-phys.net/15/2903/2015/

*Ll 651ff: It would be nice if you could give an example of estimated fluxes here as well and not only refer to other papers in the special issue.*

Adding any fluxes would require substantial additional text and ultimately the reader would need to refer to a dedicated paper. In the interest of readability and to avoid the overview paper getting longer, we would like to keep the text as it is.

*Ll 698ff: Is the coverage also too sparse for estimating CH4 fluxes? I would imagine that it is not only a matter of higher spatial resolution but also depends on the revisit time to 'see' more cloud free scenes.*

Higher spatial resolution is related to the number of cloud-free scenes, but we take your point. It is recognized that viewing the atmosphere at one local time of day will lead to a sampling bias but in practice, columns represent a superposition of emissions from of a range of times and geographical regions. Early results from TROPOMI are encouraging and could very well result in a big change in our ability to observe our small, cloudy island.

*Fig 2: It seems to be not very useful to display one-minute mean values over 4 years, the individual values are not visible at all. Maybe aggregate the measurements to lower temporal resolution or show a much shorter time period.*

We have redrawn this figure and superimposed a lower temporal resolution version of the timeseries. We believe there is value in showing also the higher-resolution data.

*Fig 9 top panel: Where does the 'outlier' with a value of âĹij23 ppm in the Jul observations at approximately 40 degrees come from?*

The outlier in this panel is at a measurement site in Romania, known as the Black

Sea, Constanta (BSC) site. This measurements of CO2 at this site do not display is significant seasonal cycle between January and July, unlike many other sites at similar latitudes. It is likely that this site is affected by local emissions that are not included in our models.

*Fig 11: The different lines in the time series plot for GOSAT are hardly visible. Tab 8: This table doesn't convey much information and could be removed.*

This figure has been redrawn with thicker lines.

*Tab 9: 'E' and 'L' stand for Eulerian and Lagrangian model type, please explain in the table caption.*

Text changed accordingly.

**Reviewer 3**

*The introduction should refer to other networks, such as the global greenhouse gas reference network from NOAA ESRL GMD, the research infrastructure ICOS, or the Integrated Global Greenhouse Gas Information System IG3IS.*

The focus of the paper is a regional network, but we have now acknowledged how GAUGE sits within larger international activities.

*In this overview paper many papers are referenced as "in preparation for ACP" (Stavert et al., Wenger et al., Connors et al., Helfter et al., Pitt et al., Lunt et al., Palmer et al.), is it the intention to have these published soon so that they can be properly referred to? This would be very helpful.*

The papers are in various stages of preparation and submission.

*Balloon CO2 sondes: The section seems a bit speculative. The two profiles (morning and afternoon) start deviating strongly (1-10 ppm) above about 4 km, while the description in the text mention problems with baseline drift and span measurement during one of the flights affecting only data collected above 6.5 km. Given that the traceability re-*

*quirements for using such data for validation of space-borne remote sensing of GHG measurements is quite tight, the balloon CO2 sonde is far away from being useful. I suggest shortening the section, just pointing out the status of the system.*

Admittedly, this is a proof-of-concept measurement system. We have now provided sufficient information (addressing the other reviewer comment) for the reader to understand the system and to appreciate its performance during GAUGE.

*Intercalibration activities: As multiple calibration scales are used for CH4 and for N2O, it should be made clear that for use in (inverse) modelling the data need to be put on the same scale to reduce any impact from bias errors.*

We clarified this point in the main text.

*Table 1: Please explain "CG-MD"*

Text changed accordingly.

*L 177: The Thoning et al. (1989) method does not provide a baseline, it is rather a curve fitting using harmonics and trend. A baseline would need to have some filtering of e.g. polluted episodes to retain the "baseline conditions".*

Lower-frequency harmonics would effectively remove high-frequency polluted events and could be used to determine baseline conditions. We have clarified this point in the main.

*Fig. 2, left panels: the colours used for the time series plots differ quite a bit from those in the legend. I suggest adopting the legend colours for the plotting.*

We have redrawn this figure using consistent colours for the data and the legend.

*L 186: the role of the boundary layer height during winter should be explained a bit more.*

Agreed. We have clarified the role of boundary layer height during winter.

*L 202: "We added an extra flask to the collection" this is unclear*

We have clarified the text.

*L 237: suggestion to replace "to" with "and"*

Typo. We have amended the text.

*L 250: "boundary layer likely plays the dominant role" do you mean boundary layer height? Otherwise this is a trivial statement as the boundary layer is sampled by the sites.*

We are missing a word. We have amended the text.

*L 286: "Differences between sailing ..." does that refer to different trips or directions of the ferry? Should this be "sailings"?*

Different between individual sailings.

*L 345: "and more sensitive" -> "and higher sensitivity" L 535: replace ";" with "and"*

This statement has several typos (also recognized by another reviewer). We have now amended the text

*L 560: "free-running CTMs" from Table 9 it seems that all transport models use assimilated meteorological fields*

This is a good point. Free-running CTMs refers to the models not being fitted to observed mole fraction data. We have clarified the text.
* * *